# Nanoparticle-Mediated Drug Delivery Systems for Precision Targeting in Oncology

**DOI:** 10.3390/ph17060677

**Published:** 2024-05-24

**Authors:** Kamelia Hristova-Panusheva, Charilaos Xenodochidis, Milena Georgieva, Natalia Krasteva

**Affiliations:** 1Institute of Biophysics and Biomedical Engineering, Bulgarian Academy of Sciences, “Acad. Georgi Bonchev” Str., Bl. 21, 1113 Sofia, Bulgaria; kamelia.t.hristova@gmail.com (K.H.-P.); xenodochidis.ch@gmail.com (C.X.); 2Institute of Molecular Biology “Acad. R. Tsanev”, Bulgarian Academy of Sciences, “Acad. Georgi Bonchev” Str., Bl. 21, 1113 Sofia, Bulgaria; milenageorgy@gmail.com

**Keywords:** nanotechnology, site-specific cancer therapy, personalized oncomedicine, cancer-cell-targeted nanomedicines, tumour microenvironment-targeted nanomedicines

## Abstract

Nanotechnology has emerged as a transformative force in oncology, facilitating advancements in site-specific cancer therapy and personalized oncomedicine. The development of nanomedicines explicitly targeted to cancer cells represents a pivotal breakthrough, allowing the development of precise interventions. These cancer-cell-targeted nanomedicines operate within the intricate milieu of the tumour microenvironment, further enhancing their therapeutic efficacy. This comprehensive review provides a contemporary perspective on precision cancer medicine and underscores the critical role of nanotechnology in advancing site-specific cancer therapy and personalized oncomedicine. It explores the categorization of nanoparticle types, distinguishing between organic and inorganic variants, and examines their significance in the targeted delivery of anticancer drugs. Current insights into the strategies for developing actively targeted nanomedicines across various cancer types are also provided, thus addressing relevant challenges associated with drug delivery barriers. Promising future directions in personalized cancer nanomedicine approaches are delivered, emphasising the imperative for continued optimization of nanocarriers in precision cancer medicine. The discussion underscores translational research’s need to enhance cancer patients’ outcomes by refining nanocarrier technologies in nanotechnology-driven, site-specific cancer therapy.

## 1. Introduction

The global impact of cancer extends beyond its immediate health implications, influencing both vulnerable populations and economies worldwide. In 2022 alone, the United States witnessed an estimated 1,918,030 new cancer cases, resulting in 609,360 deaths [1]. The burden is notably pronounced in Europe, where nearly a quarter of all cancer cases globally are reported, despite the continent constituting only one-tenth of the world’s population. Among the most lethal cancers are lung and colorectal cancers, along with breast cancers in women and prostate cancer in men (Figure 1). Lung cancer, in particular, emerges as a significant contributor, with a daily toll surpassing breast, prostate, and pancreatic cancers combined and more than double that of colorectal cancer, the second leading cause of cancer death [2,3]. According to research by the American Cancer Society, if present incidence rates do not change, it is estimated that 35 million instances of cancer will occur by 2050, primarily due to population expansion and ageing (https://pressroom.cancer.org/GlobalCancerStatistics2024, accessed on 4 April 2024). Factors, including smoking, high alcohol use, poor food, and physical inactivity, are to blame for this increase in anticipated cancer cases. These variables are also likely to increase the global cancer burden in the future. The sharpest rise is anticipated in low-income and middle-income countries, ill-equipped to meet this escalating challenge. In over 50% of countries, cancer already ranks as the first or second leading cause of death before the age of 70, a prevalence expected to intensify [4].

Beyond the profound societal impact of cancer, its economic burden is substantial, affecting patients, healthcare systems, and entire nations. The economic hardships stem from healthcare spending and productivity losses due to morbidity and premature mortality [5]. While the global financial burden remains unclear, data from selected countries illustrate the severity of the issue. In the US in 2017, cancer-related healthcare spending amounted to US$161.2 billion, with additional costs of US$30.3 billion attributed to productivity loss from morbidity and US$150.7 billion from premature mortality. This translated to an economic burden of approximately 1.8% of the GDP (gross domestic product). In the European Union, healthcare spending and productivity losses amounted to €57.3 billion, €10.6 billion, and €47.9 billion, respectively. When factoring in informal care costs of €26.1 billion, the total economic burden reached €141.8 billion, equivalent to 1.07% of GDP [5].

Here, we explore the dynamic landscape of precision cancer therapy facilitated by nanotechnology. We emphasise recent advancements in the field, encompassing targeted drug delivery systems, integrating personalized medicine approaches, and incorporating emerging strategies such as artificial intelligence (AI). Unlike previous reviews, we consolidate significant breakthroughs and forecast future developments, highlighting the critical need for interdisciplinary collaboration between medical and nanotechnology disciplines. This comprehensive approach provides a unique perspective on nanoparticle drug delivery systems in oncology, underscoring their transformative potential in cancer treatment. We critically examine current conventional therapies, assessing both their achievements and limitations. Emphasis is placed on the importance of precision cancer medicine and the role of nanomedicines in selectively targeting cancer cells and the tumour microenvironment. Thus, our review advocates for translational research to bridge the gap from bench to bedside and promotes global collaboration to refine nanocarrier technologies, ultimately offering new hope for patients and clinicians.

## 2. Current Status of Conventional Cancer Therapies: Evaluating Achievements and Delays in Fulfilling Promises

Traditional therapeutic modalities include surgical interventions, chemotherapy, and radiotherapy, with the selection of the most suitable approach contingent upon the distinct physiological characteristics of the afflicted organ. Although these therapies have achieved some clinical success, they still have many limitations that decrease their effectiveness [6,7]. Cancer classification spans over 200 types, delineated by their initiation site within the body, such as breast, lung, skin, colorectal, and liver cancers [8]. Alternatively, cancers may be categorized based on their cell or tissue origin, including carcinoma (arising in the skin or epithelial tissue lining internal organs), sarcoma (originating in connective or supportive tissues like bone, cartilage, fat, muscle, or blood vessels), leukaemia (affecting blood cells), lymphoma and myeloma (emanating from immune system cells), and brain and spinal cord cancers. The tumour stage is the main factor in the choice of cancer therapy [9]. Early detection dramatically determines the extent of a cancer patient’s survival because the last stage, IV, can be only managed but not cured.

Conventional treatments possess some severe limitations. Surgery is most effective only at an early stage of cancer disease, while radiation therapy and chemotherapeutic agents, frequently combined in patients at stages III and IV, can damage healthy cells, organs, and tissues [10,11]. Conventional chemotherapeutic agents act by destroying rapidly dividing cells, which is the leading property of neoplastic cells. However, many normal cells also divide rapidly, such as bone marrow cells, macrophages, cells in the digestive tract, and hair follicles [12]. Another disadvantage of traditional chemotherapy is its lack of specificity, which can lead to dangerous side effects such as organ failure, alopecia (hair loss), mucositis (inflammation of the digestive tract lining), myelosuppression (decreased production of white blood cells causing immunosuppression), anaemia, or thrombocytopenia. Sometimes, the adverse effects force a change in the prescribed therapy’s dosage, a postponement of treatment, or its discontinuation [12,13]. Cell division can stop close to the centre in solid tumours, rendering chemotherapy-insensitive drugs effective.

Furthermore, chemotherapeutic agents often cannot penetrate and reach the core of solid tumours, thus failing to kill the cancerous cells [14]. Traditional chemotherapeutic agents are usually cleared from the circulation and are engulfed by macrophages. Therefore, they remain in circulation for a very short time and cannot interact with the cancerous cells, making the chemotherapy ineffective. The poor solubility of the drugs is also a significant problem in conventional chemotherapy, making conventional drugs unable to penetrate the biological membranes [15]. Another major problem in chemotherapy is drug resistance, wherein cancer cells initially suppressed by an anticancer drug develop resistance. This is associated with P-glycoprotein, which is overexpressed on the surface of the cancerous cells and prevents drug accumulation inside the tumour, acting as the efflux pump and often mediating the development of resistance to anticancer drugs [16]. The limitations and drawbacks of conventional anticancer therapies are given in Figure 2.

In summary, conventional cancer therapies are integral components in the treatment paradigm, guided by the physiological characteristics and staging of the malignancy. However, these traditional approaches encounter significant limitations, including lack of specificity leading to harmful side effects, reduced effectiveness in solid tumours due to restricted penetration, and drug solubility and resistance challenges. These drawbacks highlight the need for more targeted and precise therapeutic strategies. Transitioning from the constraints of conventional treatments, the next section will delve into the realm of precision cancer medicine and targeted therapies. This emerging field aims to address the shortcomings of traditional approaches by tailoring treatments to the unique genetic and molecular characteristics of individual patients and their specific cancer types, offering a more effective and personalized approach to cancer care.

## 3. Nanotechnology in Precision Cancer Therapy

### 3.1. Precision Cancer Medicine—More Than Medicine

The concept of “precision medicine”, closely linked to personalized medicine, gained prominence around 2010. It was significantly advanced by the US National Research Council and further propelled by President Obama’s Precision Medicine Initiative (PMI) in 2015, aimed at delivering precise treatments [17]. While precision and personalized medicine are often used interchangeably, subtle distinctions exist. For instance, the US National Research Council favoured “precision medicine” for its report, suggesting nuanced distinctions in meaning. This ambiguity has sparked scholarly debate alongside related terms like “stratified medicine” and “P4 medicine” [3,18]. By leveraging comprehensive patient data, including genetic profiles and molecular characteristics, precision medicine enables clinicians to forecast disease progression and tailor preventative strategies accordingly. Moreover, it empowers clinicians to implement targeted prevention strategies based on individualized risk assessments, potentially reducing the incidence of tumour development or progression. This holistic approach extends precision medicine beyond treatment modalities, encompassing a proactive stance toward disease management and prevention [19].

Precision medicine and targeted drug delivery are complementary approaches that aim to optimize therapeutic outcomes by considering individual patient characteristics and enhancing the specificity of drug administration. It involves tailoring medical treatment based on an individual’s genetics, environment, and lifestyle, enabling the identification of the most effective therapies for specific patient subgroups [3]. On the other hand, targeted drug delivery strategies focus on modulating a drug’s pharmacokinetics and biodistribution to enhance its delivery to the disease site or target cells while minimizing off-target effects. These targeted delivery approaches, which utilize specialized drug carriers and formulations, play a crucial role in precision medicine by facilitating the administration of synergistic drug combinations and improving the therapeutic index of cancer drugs [20,21]. 

In oncology, the term precision cancer medicine describes a subset of precision medicine that focuses on individualized oncological treatment plans that integrate genetic profiles, molecular traits, and clinical data [22]. This approach enhances patient classification, drug specificity, and optimized dosing, transforming oncology practices. It promises to improve patient outcomes, reduce systemic toxicity, and overcome drug resistance (Figure 3). Exactly, the genomic analyses and molecular diagnostics form the foundation of precision cancer medicine. Analysing a tumour’s genomic profile, including mutations and gene amplifications, helps identify specific treatment options and select suitable patients for these therapies [23]. Precision cancer medicines, including antibodies and small molecules, specifically target molecules activated in cancer cells, thus disrupting tumour growth and metastasis. This data-driven approach customizes treatments based on unique patient profiles, moving away from a one-size-fits-all model and stratifying patients to determine the most effective strategies [24]. The National Cancer Institute’s (NCI) ComboMATCH initiative exemplifies precision medicine’s potential by evaluating innovative drug combinations targeting specific tumour alterations [25]. Another one example was provided by the French National Cancer Institute highlights the availability of 144 drugs, including 107 targeted therapies and 37 specific immunotherapies, primarily for patients with advanced or relapsed cancer. These tailored treatments have demonstrated promising outcomes in various cancers, such as chronic myeloid leukaemia, lung and breast cancer, and metastatic melanoma [26].

Similarly, the MOSAIC project aims to create a vast repository of spatial omics data in cancer, integrating clinical annotations with advanced profiling techniques to uncover cancer subtypes and identify drug targets and biomarkers [27]. Despite recent FDA approvals heralding a new era in precision cancer medicine, challenges remain in translating these medicines into clinical practice. Limitations such as poor targeting abilities and immunological toxicities hinder seamless integration into established therapeutic protocols [28,29]. Therefore, integrating cutting-edge technologies is essential to enhance the precision and effectiveness of these treatments. One such technology that has shown tremendous potential is nanotechnology. By offering highly selective drug delivery, responding to specific stimuli, and ensuring controlled release, nanotechnology can significantly augment the capabilities of precision cancer medicine.

### 3.2. Nanoparticles in Cancer Therapy

Nanotechnology offers a promising solution to these challenges by enabling highly selective drug delivery, responding to specific stimuli, and ensuring controlled release. Nanoparticles enhance drug stability, solubility, and retention time at tumour sites, addressing the limitations of conventional and precision medicines. Integrating nanotechnology into precision medicine represents a paradigm shift in drug delivery, potentially achieving unprecedented levels of efficacy and specificity. The convergence of initiatives like ComboMATCH, precision cancer medicines, and nanotechnology promises to reshape cancer treatment, paving the way for more effective and personalized therapeutic interventions [30,31,32]. 

In advancing nanoparticle-based drug delivery systems in oncology, it is imperative to emphasise the comparative advancements across various nanocarriers. These systems encompass diverse nanoparticles, including liposomes, polymeric nanoparticles, dendrimers, nanocrystals, nanogels, and quantum dots, offering unique advantages and applications in cancer therapy [33]. Liposomes, composed of lipid bilayers, provide controlled release and targeted delivery to tumour cells [34]. At the same time, polymeric nanoparticles, such as PLGA or PEG-based formulations, offer sustained drug release and improved bioavailability [35]. Dendrimers, with their highly branched structures, enable precise targeting of ligands to tumour sites [36], while nanocrystals enhance drug solubility, stability, and cellular uptake, thereby enhancing tumour delivery [37]. Nanogels, formed from cross-linked polymer networks, provide a controlled release and targeted drug delivery [38], while quantum dots and semiconductor nanoparticles offer unique optical properties for imaging and targeted therapy [39,40]. These nanoparticles vary in size, shape, biocompatibility, and selectivity [41,42], with most falling within the clinical range of 3 to 200 nm in diameter [42]. Over the years, various materials, including polymers, liposomes, and dendrimers, have been utilized to formulate drug carriers [42]. Among these formulations, polymer-based drug conjugates emerge as versatile constructs, leveraging both natural (e.g., albumin, chitosan, heparin) and synthetic polymers (e.g., PEG, HPMA) to encapsulate or covalently bind bioactive compounds, thereby facilitating the targeted delivery of oligonucleotides, DNA, proteins, and various pharmaceutical agents [43]. In light of these advancements, it becomes crucial to comprehensively assess these nanocarriers’ characteristics and efficacy in cancer therapy [44,45]. Evaluating and comparing the performance of these diverse nanoparticle formulations will contribute significantly to the ongoing progress in precision medicine for cancer treatment.

The attention is directed toward dissecting the multifaceted relationship between nanoparticles and cancer therapeutics, encompassing their varied classifications and regulatory approvals by the FDA. The first nanoparticle approved by the FDA in 1995 was a PEGylated liposome loaded with doxorubicin (Doxil) to treat AIDS-associated Kaposi’s sarcoma [46]. This formulation drastically reduced the side effects of doxorubicin. Since then, the FDA has approved other liposomal formulations for cancer therapy, such as Myocet and DaunoXome (Table 1) [47,48,49]. Although liposomes and micelles are two or more phospholipid-based layer vesicles, their morphologies differ. They are mainly employed to encapsulate hydrophilic medicines in their aqueous core and have a structure similar to cell membranes. Hydrophobic drugs can be chemically bonded to liposomal particles or accommodated in the bilayer [50]. Hydrophobic medications can be encapsulated in micelles thanks to their hydrophobic core [51]. The FDA has already approved some formulations based on polymeric nanoparticles, such as Abraxane (albumin–paclitaxel particles for the treatment of pancreatic ductal adenocarcinoma and metastatic breast cancer) and Ontak (an engineered protein combining diphtheria toxins and interleukin-2 for the treatment of non-Hodgkin’s peripheral T-cell lymphomas) [52]. They demonstrated reduced loading capacity and systemic toxicity but increased biocompatibility and biodegradability [53,54,55]. 

In light of these considerations, alternative avenues in nanoparticle design and formulation, particularly concerning the balance between biocompatibility, loading capacity, and systemic toxicity, started to appear. As such, the inorganic NPs, another class of NPs, are characterised by high stability and low biodegradability. They include carbon-based quantum dots and hybrid, silica and polystyrene, and metallic NPs and are generally used as contrast agents for diagnosis [54]. However, combined with active ingredients, they can be promising tools for theranostic applications. As an illustration, superparamagnetic iron oxide nanoparticles (SPIONs) are commonly utilized as contrast agents in magnetic resonance imaging (MRI) owing to their magnetic responsiveness [56]. Polymeric iron oxide nanoparticles (PIONs) have also garnered attention for their potential in cancer therapy through magnetic hyperthermia. Notably, a specific formulation known as Nanotherm, consisting of iron oxide coated with amino silane, has received approval to treat glioblastoma [57]. The other inorganic NPs are quantum dots, tiny semiconductor nanocrystals that emit light and have remarkable optical and electrical characteristics that make them sensitive, photobleach-resistant, and highly fluorescent. Their primary applications have been in imaging and detection. They were coupled to anti-HER2 antibodies and coated with poly (ethylene glycol) (PEG) in recent work to enable localization in specific tumour cells [58].

Another illustration is the range of gold nanoparticles that, due to their optical and electrical characteristics and low toxicity [59,60,61], are employed as contrast agents for computed tomography [62], photoacoustic imaging [63], and photodynamic treatment [64]. The PEG-coated gold shell of the nanoshell-based AuroShell comprises a silica core. It was brought to market as AuroShell (Nanospectra) after receiving FDA approval in 2012 to treat breast cancer using photodynamic therapy [62]. 

**Table 1 pharmaceuticals-17-00677-t001:** FDA-approved nanomedicines based on nanoparticles for cancer therapy.

Drug Name	Type NPs	Active Drug	Type of Cancer	Ref.
Doxil	Liposomal	Doxorubicin	Ovarian cancer, AIDS-related Kaposi’s sarcoma/Multiple myeloma	[65]
Myocet	Liposomal	Doxorubicin	Metastatic breast cancer	[66]
Onivyde	Liposomal	Irinotecan	Metastatic pancreatic cancer	[67]
Daunoxome	Liposomal	Daunorubicin	HIV-associated Kaposi’s sarcoma (KS.)	[68]
Vyxeos	Liposomal	Daunorubicinand Cytarabine	Acute myeloid leukaemia	[69]
Ameluz	Liposomal	5-aminolaevulinic acid	Actinic keratoses	[70]
Abraxane	Polymeric	Paclitaxel	Breast cancer, Non-small cell lung cancer and Pancreatic cancer	[71]
Genexol-PM	Polymeric	Paclitaxel	Breast Cancer, Non-small cell lung cancer and Ovarian cancer	[72]
NKTR-102	Polymeric	Irinotecan	Breast cancer, Ovarian and Colorectal cancer	[73]
Opaxio	Polymeric	Paclitaxel	Lung cancer, Ovarian and Cervical cancers	[74]

Apart from the drug delivery systems which have been either accepted or are under clinical investigation, there are new nanoparticles currently under research that could improve treatment performance. Examples are lipid nanoparticles loaded with SPIONs and temozolomide, combining the effect of conventional chemotherapy and hyperthermia to treat glioblastoma [75,76]. Lipid nanoparticles are good candidates for brain tumour therapy as they can cross the blood–brain barrier (BBB) [77]. Another class of polymeric nanoparticles is called dendrimers, which are distinguished by having a spherical shape and a repeating branching structure [78,79]. Their structure is incredibly adaptable for various applications since their design is easily controllable. For instance, some recent research demonstrates that in in vivo tumour models, poly-L-lysine (PLL) dendrimers loaded with doxorubicin elicit anti-angiogenic responses [80]. Currently, there is just one clinical study for ImDendrim, a formulation for treating inoperable liver tumours that do not respond to standard therapy. ImDendrim is based on a dendrimer and rhenium complex connected to an imidazolium ligand [81].

Equally important is the contribution of biomimetic nano-systems in cancer therapy. The reason lies behind their advantageous properties as they improve targeting nanocarriers’ biocompatibility and drastically decrease their biotoxicity [82]. A growth in protein- and cell membrane-based biomimetic nanocarriers implementation has been observed in treating different tumour types. 

Surface-modified nanocarriers cannot be excluded from the long list of applied NPs. Over the years, various methods have been developed, including applying polymers, surfactants, ligands and fatty acids, optimizing parameters such as controlled release, precise drug delivery, etc. In short, polymers like polyethylene glycol (PEG) and chitosan have improved the efficiency of lipid-based nanocarriers via the enhanced permeability and retention effect. Also, fragment antigen binding has been considered a big player in delivering anticancer drugs to targeted cells. Another example of ligands is aptamers loaded onto the surface of lipid-based nanoparticles targeting breast cancer [83]. Similarly, authors have reported promising cancer detection and therapy results after using surface modifiers such as folic, lauric, and myristic acids [84]. In this case, the acids provided chemical groups that suggested nanoparticles firmly bond with various cancer cells. 

### 3.3. Nanoparticles for Targeted Drug Delivery in Precision Cancer Medicine

#### 3.3.1. Molecular and Ligand-Based Targeting Nanoparticles for Precision Cancer Therapy 

Due to the advanced understanding of tumour biology, researchers have exploited different cancer cell surface proteins, dysregulated oncogenes, and signalling pathways as potential targets for selectively delivering drugs [85]. Given the critical need for more effective drug delivery systems, nanotechnology offers a promising solution, potentially revolutionizing the field. 

Figure 4 illustrates the two primary mechanisms through which nanoparticles (NPs) achieve targeted delivery in oncology: passive and active targeting. Passive targeting relies on the enhanced permeability and retention (EPR) effect, taking advantage of leaky vasculature and impaired lymphatic drainage in tumours. Active targeting involves the functionalization of NPs with ligands that specifically bind to receptors overexpressed on cancer cells, facilitating precise delivery. This visual representation elucidates the crucial role of NPs in advancing precision medicine approaches for cancer therapy. Several targeting ligands, such as aptamers, small molecules, and antibody fragments, incorporated in NPs bind to overexpressed receptors on cancer cells, increase tumour specificity, and reduce off-target drug toxicities [86]. Moreover, the conjugation of NPs with cancer cell-targeting ligands improves internalisation and receptor-mediated endocytosis [87,88]. 

Because of their facile accessibility, high attachment affinity, and specificity for targeting cancer cells (e.g., breast cancer), antibodies are now the most utilized ligands for active targeting. The characteristic Y” form of these proteins indicates that their two arms work together to bind with the antigen [89] selectively. Small peptides and proteins are other types of ligands, and they have several benefits, such as a smaller molecular weight, the ability to diffuse molecules, the reduction of immunogenicity, simplicity of production, and relative flexibility in chemical conjugation techniques [90]. Small synthetic single-stranded RNA or DNA oligonucleotides called aptamers are folded into certain forms that enable them to attach to particular targets [91]. They can produce complex three-dimensional structures that cling securely and highly selectively to surface markers [92]. Precision medicine has led to the development of cancer drugs targeting specific antigens associated with tumours. Several clinical trials are currently testing ligand-targeted nanoparticles as potential cancer treatments, as summarized in Table 2.

Identifying tumour-associated antigens (TAA) has made the development of antigen-specific cancer therapies for precision medicine possible. Several ligand-targeted nanoparticles are in clinical trials for cancer therapy. Some are summarized in Table 2.

The prostate-specific membrane antigen (PSMA) is specific for the prostate, and its levels are overexpressed in prostate cancer plasma. They thus are exploited to deliver medicines to prostate tumours [93] preferentially. Consequently, efforts have been directed towards leveraging PSMA to deliver therapeutics to prostate tumours preferentially. For instance, PSMA-targeted polymeric nanoparticles loaded with docetaxel (BIND-014) underwent phase I and II studies, demonstrating potential in selectively targeting metastatic prostate tumour cells overexpressing the PSMA receptor [94]. BIND-014 was retained in the vascular compartment in preclinical toxicokinetic studies conducted in mice, rats, and monkeys and the phase I clinical trial (NCT01300533). This was related to improved efficacy in various cancer types, such as cervical and cholangiocarcinoma, and suggested altered pharmacokinetics of the particles [94,95]. The results further indicated increased uptake of the nanoparticles by the cancer cells/neovasculature due to the presence of the PSMA-targeting ligand on the nanoparticles and increased accumulation of the nanoparticles through the enhanced permeation and retention (EPR) effect [94]. BIND-014 showed anti-tumour activity in three patients with metastatic castration-resistant prostate cancer who had not received chemotherapy in the next phase II clinical trial (NCT01812746). It also increased the median overall survival time (13.4 months) and reduced PSA levels by 50% in 30% of patients eligible for PSA evaluation [94,95].

In pursuing tailored therapeutic interventions, the strategic utilization of specific biomarkers, such as PSMA in prostate cancer and HER2 (human epidermal growth factor receptor 2) in breast cancer, underscores a paradigm shift towards precision medicine approaches. Higher levels of HER2 characterise breast cancer, and to target breast cancer cells, *HER2* ligands are conjugated to different types of nanoparticles (NPs). In preclinical and early phase I clinical trials, MM-302 alone or combined with trastuzumab (which binds to a different HER2 epitope) have shown an improved efficacy and safety profile in metastatic breast cancer patients [96]. These NPs have been directed to phase II of a clinical trial (HERMIONE; NCT02213744) but did not show a clear benefit in terms of survival rate compared to conventional chemotherapies. Other specific targeted receptors expressed in breast cancer are the transferrin receptor (TfR), epidermal growth factor receptor (EGFR), and folate receptor. Human breast cancer cells (MCF-7) showed increased cellular absorption of rapamycin–PLGA nanoparticles coupled to EGFR antibodies and more significant apoptotic activity [97]. Human serum albumin NPs loaded with loperamide and coupled to antibodies targeting transferrin receptors could pass the blood–brain barrier and deliver the medication to the intended location [98]. The last NPs, however, still have not reached the pre- and clinical trials phases.

Increased levels of the epidermal growth factor receptor (EGFR) are also observed in non-small cell lung carcinoma (NSCLC) cells (40–80%) [99]. Mucin 1 (MUC1) is a cell membrane glycoprotein overexpressed in breast and lung cancer. Only a few nanoparticles designed to target lung cancer are currently under clinical trials [100]. Among them, tecemotide is a synthetic lipopeptide that targets tumour-associated MUC1, expressed by over 90% of lung cancers. It is being evaluated in Phase III clinical trials for the treatment of metastatic stage IIIA/IIIB non-small cell lung cancer as maintenance therapy following chemoradiotherapy. It is well tolerated by the human body but has demonstrated low or insignificant efficacy [101,102]. The surface of gelatin carriers was functionalized with biotinylated EGF for targeted cisplatin delivery to metastatic lung cancer cells. As a result, reduced kidney toxicity in mice was found [103]. Luteinizing hormone-releasing hormone receptors (LHRHR) are also overexpressed in lung cancer cells. Mesoporous silica nanocarriers targeting LHRHR have been shown in another investigation to effectively deliver anticancer payloads into lung cancer cells. LHRH peptide was used to functionalize the surface of the nanocarriers, and Dox, cis-platin, and siRNA were added. Mesoporous silica nanocarriers loaded with cisplatin or LHRH-PEGylated Dox exhibited improved cytotoxicity because of their accumulation in lung cancer cells [104]. Polymer-based nanoparticles, as well as lipid-polymer hybrid nanoparticles loaded with cisplatin, are conjugated with hyaluronic acid (HA) and have been found to favour lung cancer cell recognition, facilitated by binding to the CD44 cell marker [105,106].

As the search for targeted therapies gains momentum, TRAIL (tumour necrosis factor-related apoptosis-inducing ligand) emerges as a promising molecule, exploiting the selective interaction with death receptors such as DR4/TRAIL-R1, notably overexpressed on cancer cells. This potential was highlighted in a study by Kim et al. (2013), wherein porous PLGA carriers incorporating Apo2L/TRAIL facilitated the specific delivery of Dox to metastatic lung cancer cells, with sustained accumulation in the mouse lung following pulmonary administration [107]. This delivery system exhibited significant apoptosis, indicating synergistic cytotoxicity when H226 cells (a human cancer cell line derived explicitly from lung cancer) were co-treated with TRAIL and Dox. To overcome the limitations of tumour penetration, a tumour-homing peptide known as iRGD (CRGDKGPDC), targeting αv integrins on lung cancer cells, was used. Acetylated dextran nanocarriers were engineered with iRGD and loaded with paclitaxel, resulting in inhalable dry powder nanocomposites for the targeted lung cancer treatment [108].

As investigations into targeted therapies for lung cancer advance, alongside the promising utilization of TRAIL-based delivery systems, recent endeavours have also sought to capitalise on specific molecular targets like the folate receptor-alpha overexpressed on lung tumour cells. Polyethylene glycol and chitosan copolymer nanocarriers loaded with paclitaxel and folate-grafted were designed by Rosiere et al. [109] to target specifically the folate receptor-alpha overexpressed lung tumour cells. In healthy mice, the release profile was retained with prolonged retention of paclitaxel of around seven hours inside the lungs. Other examples are the EGF-functionalized PLGA NPs loaded with 5-FU and perfluorocarbons, which have been shown to inhibit colon tumour growth [110]. Composite PLGA/PLA-PEG-FA NPs were designed to incorporate 17-AAG (NP-PEG-FA/17-AAG) EGF-functionalized PLGA nanoparticles loaded with 5-FU, and perfluorocarbons demonstrated suppression of colon tumour growth [111,112]. A composite formulation of PLGA/PLA-PEG-FA NPs was engineered by incorporating 17-AAG and targeting the folic acid receptor [113]. This formulation, NP-PEG-FA/17-AAG, improved the oral bioavailability of 17-AAG and proved effective in treating ulcerative colitis and associated cancer [113]. PHBV/PLGA NPs loaded with 5-FU exhibited promise as a nano-drug delivery system for colon cancer treatment [114,115,116].

The examples above of actively targeted nanoparticles underscore promising strides towards achieving tumour selectivity within clinical contexts. Yet, a more profound comprehension of cancer heterogeneity and variations in receptor expression across primary and metastatic sites is imperative to enhance the efficacy of ligand-targeted nanoparticle delivery systems. Such insights are pivotal for refining receptor-targeting strategies across different stages of cancer progression. Furthermore, as research continues to unravel the complexities of nanoparticle interactions within the tumour milieu, insights pave the way for integrating novel strategies targeting the tumour microenvironment, thus shaping the trajectory of future advancements in cancer nanomedicine. 

#### 3.3.2. Nanoparticles in Integrated Diagnostics and Therapeutics for Cancer 

Nanotheranostics emerges as a promising cancer diagnosis and treatment frontier, leveraging nanotechnology for simultaneous imaging and therapy. It combines targeted therapy based on specific diagnostics and holds great potential in personalized cancer nanomedicine (PM) [117,118]. Theranostics could be used in many areas of personalized treatment, including early disease detection, disease staging, therapy selection, treatment planning, early detection of side effects, and follow-up therapy planning by fusing molecular imaging and molecular therapy. An ideal PM theranostic system for cancer would identify the disease’s class, scan the tumour’s heterogeneity, administer a customized treatment based on the findings of the diagnostic and imaging processes, and then track the effectiveness of the therapy. Nanotechnologies play a significant role in theranostics. The use of nanoparticles in medicine offers several benefits for both diagnosis and therapy, which is why nanomedicine emerged. The latter can use receptor-mediated active targeting or extravasation from blood arteries into the tumour site to deliver medications at larger dosages with fewer adverse effects [119]. Nanocarriers can also integrate imaging agents to monitor drug delivery, tumour targeting, and treatment response. This enables clinicians to tailor treatment strategies based on individual patients’ responses, improving outcomes and reducing side effects. For instance, silica nanoparticles (C-dots) labelled with dual 124 I PET and Cy5 optical imaging agents and with cRGD-targeted ligands [94] were found to be stable and well tolerated in patients with metastatic melanoma. Systemic administration of 124 I-cRGDY-PEG-C-dots did not result in accumulation in off-target organs such as the liver, lung, spleen, bone marrow, and gastrointestinal tracts. These nanoparticles showed no toxic or adverse events and are enabled for rapid renal clearance because of their ultra-small size [120].

#### 3.3.3. Targeting Tumour Vasculature with Nanoparticles 

The vascular structure of tumours significantly differs from that of normal tissues. Tumour vessels are chaotic and irregular, with inconsistent diameters, frequent blind ends, and a lack of hierarchical organization. Single tumour vessels also display abnormalities such as irregular endothelial cells forming weak junctions between them, poor connections between endothelial cells and pericytes, abnormal basement membranes, and deficient coverage of vascular smooth muscle cells [121]. Due to the leaky vessels, interstitial fluid pressure is high, resulting in inefficient circulation. These fundamental differences between tumour and normal tissue vasculature allow several small molecules, vascular disrupting agents (VDAs), to reduce tumour circulation. VDAs selectively target the abnormal vessels, disrupting by two main mechanisms: a direct effect on endothelial cell cytoskeleton by tubulin-binding agents, leading to vessel destabilisation, increased vessel leakage and vascular resistance, stagnation of flow, stacking of red blood cells, and vessel occlusion in tumours; induction of endothelial cell apoptosis by flavonoids, probably by the activating of tumour necrosis factor-alpha and recruitment of activated neutrophils [122,123]. It is crucial to highlight that cytoskeletal disruption targets immature endothelial cells selectively, especially those devoid of smooth muscle and pericyte contact. As a result, the tumour neovasculature is considerably more sensitive to tubulin-binding VDAs than vessels in normal tissues. Among tubulin-binding VDAs are those of BNC105P, ombrabulin, and combretastatin A4-phosphate (CA4P), which have been well studied for treating ovarium cancer (OC). 

Preclinical studies demonstrate VDAs’ ability to reduce tumour blood flow, resulting in extensive necrosis, particularly notable with combretastatin A4-phosphate (CA4P). However, tumour regrowth occurs due to surviving cells supplied by normal vasculature. Combining VDAs with chemotherapy or anti-angiogenic agents (AAs) can prevent this regrowth, targeting different aspects of tumour vasculature. Ongoing trials like PAZOFOS and FOCUS evaluate the effects of combining CA4P with AAs and chemotherapy in platinum-resistant OC. AAs are a distinct class of vascular-targeted agents that suppress the growth of new tumour vasculature through inhibition of vascular endothelial growth factor (VEGF) and other pro-angiogenic molecules (e.g., bFGF, PDGF), unlike those that target the established tumour vascular network. AAs remodel the tumour vasculature, improve vessel perfusion, and increase drug delivery to the tumour [124]. Patients with glioblastoma and rectal cancer treated with anti-angiogenic agents have shown normalization of tumour vessels, decreased vessel size and leakiness, and increased drug delivery [125,126]. Anti-angiogenic therapies have been shown to improve with greater effectiveness the delivery of nanoparticles with particle sizes of approx. 10–20 nm compared to those bigger than 100 nm [127,128].

#### 3.3.4. Nanoparticles Targeting the Tumour Microenvironment Support Systems 

Precision medicines for cancer patients, such as those mentioned above, are aimed to target mainly oncogenes or signalling pathways involved in cancer progression. The tumour microenvironment (TME) plays a central role in cancer development and has the potential to influence the response to treatments. However, characterisation of a patient’s TME is an often-neglected approach in precision oncomedicine. TME is the tumour surrounding it, which comprises several types of cells, such as immune cells, fibroblasts, endothelial cells, extracellular matrix and soluble factors, and blood vessels [129]. TME exhibits a high heterogeneity concerning cancer type, patient genetics, pathophysiological barrier, etc. For example, Job et al. have analysed the gene expression profile and cell composition of TME of intrahepatic cholangiocarcinoma and have identified four different TME subtypes based on different cell compositions and distinct mechanisms of immune dysfunction [130]. Therefore, a thorough screening of a patient’s TME could help in a more precise selection of appropriate therapies and improve therapeutic effect. Targeting TME with therapeutic strategies targeting cancer cells is a promising approach for cancer treatments with a synergistic effect [131]. Furthermore, nanoparticles utilized for drug delivery hold the potential to be meticulously engineered, enabling the preferential delivery of precision medicines to the specific and intricate landscape of the tumour microenvironment while sparing normal tissues from unintended effects. This precise targeting capability represents a critical advancement in cancer therapy, offering the prospect of heightened therapeutic efficacy with minimized off-target toxicity. Through strategic design and functionalization, nanoparticles can navigate the complex physiological barriers of tumours, facilitating the accumulation and controlled release of therapeutic payloads within the tumour microenvironment [132]. This targeted approach enhances treatment efficacy and mitigates systemic side effects, thus advancing the paradigm of personalized cancer therapy towards improved clinical outcomes.

The inherent heterogeneity characterising the tumour microenvironment (TME) across a patient’s primary tumour and between primary and metastatic sites constitutes a critical determinant influencing the efficacy of drug delivery and subsequent treatment outcomes [133,134]. 

Another aspect of TME that should be considered to enhance the delivery of precision medicine is tumour stroma [135]. The tumour stroma, or the microenvironment surrounding solid tumours, encompasses the extracellular matrix and specialized connective tissue cells, including fibroblasts and mesenchymal stromal cells. The desmoplastic stroma within tumours makes a pathobiological barrier that hampers the accumulation of large nanoparticles [136]. However, the extravasation of smaller nanoparticles may still be permitted. Various approaches have been used to diminish solid stress in fibrotic tumours and, consequently, the decompression of vessel compression. Drugs like losartan, PEGylated hyaluronidase, and inhibitors of Hedgehog signalling pathways have demonstrated promise in relieving solid stress and enhancing the accumulation of nanoparticles [136,137,138].

Furthermore, nanomedicine, which can improve the delivery and retention of drugs in the tumour, has been proposed to target the stroma [139].- Among the various TME modulators, stromal modulators (e.g., TGF-β inhibitors, hedgehog inhibitors, among others) have demonstrated more significant (>70%) tumour inhibition and have advanced to clinical trials for further evaluation [140]. Notably, several U.S. Food and Drug Administration (FDA) approved drugs, including losartan, GDC-0449 (vismodegib), and arsenic trioxide, initially indicated for hypertension, basal cell carcinoma, and refractory or relapsed acute promyelocytic leukaemia, respectively, have demonstrated improved anticancer efficacy when delivered in combination with TME modulators [141,142,143,144,145] 

In a small randomised clinical trial, the Hedgehog signalling inhibitor vismodegib, combined with gemcitabine and nab-paclitaxel, showed an improved survival rate in patients with pancreatic cancer. This highlights the potential of normalizing the tumour stroma and reducing solid stress to enhance precision nanomedicine delivery [146]. In an orthotopic pancreatic cancer model, administration of losartan (an FDA-approved anti-hypertensive drug with anti-fibrotic properties) led to increased accumulation of PEGylated liposomal doxorubicin (Doxil) compared to animals receiving Doxil alone [136]. This enhanced transport of larger nanoparticles, such as 100 nm Doxil, was attributed to reduced solid stress and vessel compression in the tumour stroma, resulting in improved vessel permeability [136]. These examples show the importance of optimizing the physicochemical properties of nanoparticles and the tumour microenvironment to translate precision nanomedicine in clinical settings successfully.

Table 3 provides a comprehensive summary of targeted formulations currently under clinical trials, categorized into four main cancer therapy strategies: Molecular and Ligand-Based Targeting Strategies for Precision Cancer Therapy, Vascular Targeting, Theranostics, and Exploiting the Tumour Microenvironment. Each formulation is listed along with its target, clinical trial phase, cancer type, and critical results. This table offers a concise overview of ongoing research efforts in developing precise and effective cancer therapies tailored to specific molecular targets, tumour vasculature, diagnostic and therapeutic integration, and the tumour microenvironment.

## 4. Future Directions in Personalized Nanomedicine Approaches

Personalized nanomedicine approaches are intricately intertwined with the evolving landscape of precision oncology, representing a promising frontier for enhanced therapeutic efficacy and targeted drug delivery systems. As scientific and technological advancements propel our comprehension of cancer pathophysiology forward, novel avenues in personalized medicine continue to emerge, offering tantalizing prospects for optimized treatment outcomes. 

Below, we delineate several of these prospective directions poised to shape the future trajectory of cancer therapeutics (Figure 5).

### 4.1. Integration of Multi-Omics Data

Omics procedures thoroughly evaluate many types of biological molecules, such as DNA, proteins, RNA, or metabolites. These approaches include proteomics, genomics, metabolomics, epigenomics, and transcriptomics. The use of individual omics techniques, such as the genetic sequencing of malignancies, has grown in clinical practice and has significantly aided in the diagnosis of diseases as well as the discovery of biomarkers that may be used to monitor the progression of illness and suggest successful therapies [147,148,149,150]. Nevertheless, the intricacy and interconnections of molecular processes are too great for an individual technique to capture. For instance, millions of risk loci for various illnesses have been found by genome-wide association studies (GWAS). However, the practical value of such discoveries is limited since the causative gene is frequently not identified [151]. 

As genomics is developing, offering solutions and at the same time raising questions, the field of epigenomics also starts taking a pivotal role in the realm of personalized oncology, offering profound insights into the intricate interplay between genetics and the environment in cancer development and progression. By scrutinizing the modifications to DNA and histones that regulate gene expression without altering the underlying genetic code, epigenomics unveils the dynamic landscape of epigenetic alterations across different cancer types and individual patients [152]. These alterations serve as potential biomarkers for early detection, prognosis, and therapeutic response prediction and provide novel therapeutic targets for precision medicine approaches. Moreover, epigenomic profiling enables the classification of tumours into distinct molecular subtypes, facilitating treatment strategies tailored to individual patients’ unique epigenetic profiles. Integrating epigenomic data with other omics technologies, such as genomics and transcriptomics, holds promise for unravelling the complexity of cancer heterogeneity and guiding personalized therapeutic interventions that maximize efficacy while minimizing adverse effects [153]. Transcriptomics and proteomics offer functional information that genomics alone cannot capture, opening up new possibilities for comprehending the molecular complexity that underlies illness. Recently, the power of separate data types has been combined to create multi-omics, a unique integration of distinct omics data that captures the intricate molecular interaction of health and illness [154]. This development was made possible by advancements in different omics technologies and processing capabilities. They support the discovery of new, more individualized targets for nanomedicine treatments.

Due to several key factors, integrating multi-omics data in precision oncology necessitates advanced computational tools and artificial intelligence (AI) techniques. Multi-omics datasets require sophisticated algorithms for data integration, dimensionality reduction, and feature selection. AI-driven predictive models leverage this integrated data to stratify patients, predict treatment responses, and identify biomarkers for personalized therapies. Furthermore, AI-powered network analysis tools uncover complex molecular interactions and pathways dysregulated in cancer, aiding in identifying therapeutic targets [155]. Real-time data processing and decision support systems enable clinicians to rapidly analyse patient-specific molecular profiles and guide personalized treatment strategies, enhancing patient outcomes in precision oncology [156].

### 4.2. Integration of Multifunctional Nanoparticles in Combined Therapeutic Systems

Advancements in nanotechnology have paved the way for the development of multifunctional nanoparticles that can revolutionize combined therapeutic systems. These nanoparticles serve as a unified platform, integrating various treatment modalities such as chemotherapy, immunotherapy, and gene therapy. Nanocarriers can be engineered to deliver simultaneously multiple drugs or therapeutic agents and thus may synchronize the biodistribution and pharmacokinetics of different medications, optimizing synergistic effects and minimizing adverse interactions [157]. Moreover, treatments can be tailored precisely by engineering “smart” nanomaterials, including stimuli-responsive nanocarriers. These nanocarriers can release drugs or genes in response to specific triggers, either endogenous (pH, enzymes, temperature, redox values, hypoxia, glucose levels) or exogenous (light, magnetism, ultrasound, electrical pulses). This controlled release mechanism ensures optimal biodistribution and targeted delivery to specific sites within the body. For instance, nanoscale materials like graphene, quantum dots, or metal–organic frameworks may offer novel properties and functionalities for targeted drug delivery [158].

Additionally, organelle-targeted therapy presents a promising frontier in cancer treatment, where treatments can be directed precisely to cellular structures involved in disease progression. While achieving controlled delivery at the organelle level is within reach, harnessing its full potential in clinical practice remains a challenge yet to be fully realised. Nonetheless, these advancements lay the groundwork for personalized cancer treatments tailored to individual patient needs.

#### Intersection of G Quadruplexes and Nanoparticles in Targeted Cancer Therapy

In the realm of cancer therapy, G quadruplexes have emerged as promising targets for intervention due to their unique structural features and functional implications in cancer biology [159]. These structures have garnered attention as potential therapeutic targets for cancer treatment, with efforts focused on disrupting their stability or exploiting their regulatory roles in cancer cell metabolism and gene expression pathways [160]. Interestingly, nanoparticles offer a compelling platform for the targeted delivery of G quadruplex-targeting agents, such as small molecules or nucleic acid-based therapeutics, to cancer cells [161]. By leveraging the specificity and versatility of nanoparticle-based delivery systems, researchers aim to enhance the efficacy and selectivity of G quadruplex-targeted therapies while minimizing off-target effects. Furthermore, integrating G quadruplex-targeting strategies into personalized nanomedicine approaches holds promise for tailored cancer therapies. By leveraging the unique molecular signatures of individual tumours, such as variations in G quadruplex expression and stability, nanomedicine can be customized to optimize treatment outcomes. This personalized approach enables the precise targeting of cancer cells while minimizing off-target effects, thus enhancing therapeutic efficacy and improving patient outcomes [158]. Thus, the convergence of G quadruplex-targeted therapy with nanoparticle-based drug delivery represents a novel avenue for advancing precision cancer therapy towards improved clinical outcomes.

### 4.3. Integration of AI-Driven Nanoparticles: Advancing Precision Medicine in Cancer Therapy

The advancement of artificial intelligence (AI) paves the way for the construction of intelligent nanoparticles that offer advancements in precision medicine. AI is a branch of computer science that performs complex tasks requiring “human intelligence” using computers or computer-controlled machines. AI algorithms aid in designing intelligent nanoparticles to overcome the limitations of conventional drug delivery systems and enhance cancer treatment efficacy. They optimize various aspects of nanoparticle design, including size, charge, drug encapsulation efficiency, interactions with biological membranes, and drug release kinetics. Computer-aided nanoparticle design encompasses various theoretical and computational methodologies such as molecular docking, dynamics, quantitative structure–activity relationships (QSAR), cheminformatics, theoretical chemistry, and similarity searching. Although these methodologies have been utilized for several years, they are continuously being developed and improved. The development of computer-aided nanoparticle design is being propelled forward by some cutting-edge ideas and methods now being explored. Examples of the second category include “big data,” artificial intelligence, machine learning, and deep learning. In drug discovery, every new and traditional method is employed with emerging trends like polypharmacology and drug repurposing. In the same way as other multidisciplinary methods, computer-aided nanoparticle design faces many challenges. These challenges include the refinement of the theoretical basis, the rational application of the technology (while being aware of its limitations), and the education and training of those utilizing it.

AI-driven nanoparticles can revolutionize cancer drug delivery, early screening, diagnosis, and predicting drug dosage and efficacy. They can accurately analyse tumour biomarkers to identify suitable patients and navigate intelligently through the tumour microenvironment to target cancer cells and selectively deliver therapeutic payloads. AI-driven nano-particles can predict nanoparticle permeability in tumour vasculatures and assist nanoparticles in entering solid tumours. Furthermore, AI-powered nanoparticles enhance therapeutic efficacy by integrating sensors and imaging agents through real-time monitoring and feedback. This closed-loop feedback system maximizes therapeutic response, overcomes drug resistance, and facilitates adaptive treatment strategies.

AI and nanotechnology can also be used for early cancer screening and diagnosis. However, it is crucial to note that the application of AI-powered nanoparticles in cancer therapy is still in its early stages. Challenges such as ensuring nanoparticle safety and biocompatibility, addressing regulatory concerns, and validating clinical efficacy through rigorous studies and trials must be addressed.

### 4.4. Clinical Translation and Regulatory Considerations

Personalized nanomedicine techniques are moving closer to being integrated into clinical practice. Thus, solving several ethical and regulatory issues is critical to guarantee their safe and efficient application. One crucial component is creating reliable manufacturing procedures for producing reproducibly high-quality nanotheranostic agents [117,118]. Standardizing manufacturing processes guarantees that the final nanoparticles satisfy strict clinical application regulatory standards while streamlining production. Apart from production, implementing uniform quality control protocols is vital to ensure the security and effectiveness of customized nanomedicine merchandise. Comprehensive characterisation methods for evaluating the stability, biological interactions, and physicochemical characteristics of nanotheranostic agents should be part of quality control measures. Strict quality control methods reduce the danger and variability in nanomaterial qualities [162].

Furthermore, the effective integration of nanotheranostics into therapeutic applications depends on resolving safety problems related to the technology. To identify possible dangers and improve dosage regimens, thorough preclinical studies of nanomaterials’ pharmacokinetics, biodistribution, and toxicity profiles are crucial [163]. In addition, careful observation of adverse events and patient outcomes throughout clinical trials facilitates early identification of safety problems and helps develop risk-reduction plans. When integrating individualized nanomedicine into clinical practice, ethical issues are far more important than technological ones. Ethical requirements include ensuring patient privacy, obtaining informed permission, and providing equal access to nanotheranostic devices. Patients are better equipped to make decisions about their care when the advantages, dangers, and uncertainties of tailored nanomedicine are openly disclosed [164].

## 5. Conclusions

In conclusion, this review underscores the transformative potential of personalized nanomedicine in revolutionizing cancer therapy. By integrating a comprehensive understanding of cancer cell-specific surface proteins, dysregulated oncogenes, signalling pathways, and the complexities of the tumour microenvironment (TME), novel avenues for optimizing nanoparticle properties and modulating the tumour stroma effectively emerge. This holistic approach lays the foundation for advancing personalized cancer nanomedicine and promises tailored therapies characterised by heightened precision and efficacy.

By highlighting the convergence of precision medicine, nanotechnology, and emerging strategies like artificial intelligence, this review consolidates recent advancements and forecasts future breakthroughs in the field. The critical emphasis on interdisciplinary collaboration between medical and nanotechnology disciplines underscores the importance of a comprehensive approach to nanoparticle drug delivery systems in oncology, emphasising their pivotal role in transforming cancer treatment. The novelty lies in the exploration of cutting-edge developments in personalized cancer nanomedicine, emphasising the integration of multi-omics data, biomarker-guided strategic nanocarrier design, nanotheranostics applications, targeted modulation of the TME, novel combination therapies, sophisticated drug delivery systems, and the translation of these innovations into clinical practice. These endeavours represent a paradigm shift in cancer care, offering the potential for customized treatments based on individual molecular profiles, thereby significantly improving clinical outcomes and reshaping the landscape of cancer therapy.

Through advocating for translational research, bridging the bench-to-bedside gap, and fostering global cooperation to refine nanocarrier technologies, we envision a future where personalized approaches in nanooncology become a game-changer in the fight against cancer. The sustained focus on research, innovation, and collaboration offers hope for improved patient outcomes and a more promising future in cancer therapy, marking the dawn of a new era where patients benefit from precisely tailored treatments based on their unique biological makeup with unprecedented accuracy and efficacy.

## Figures and Tables

**Figure 1 pharmaceuticals-17-00677-f001:**
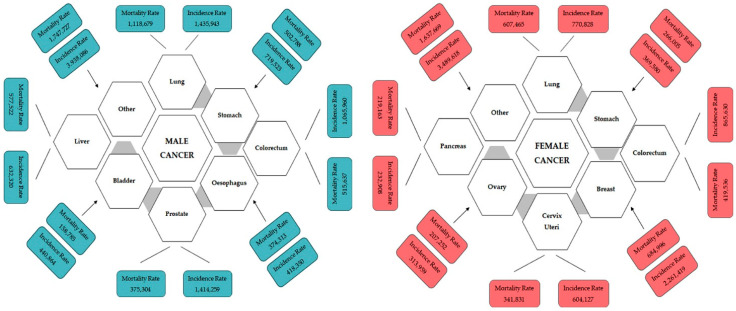
Incidence and mortality of different types of male and female cancer worldwide for 2020 estimated by GLOBOCAN 2020. The image is adapted from https://www.uicc.org/news/globocan-2020-new-global-cancer-data, accessed on 15 December 2023.

**Figure 2 pharmaceuticals-17-00677-f002:**
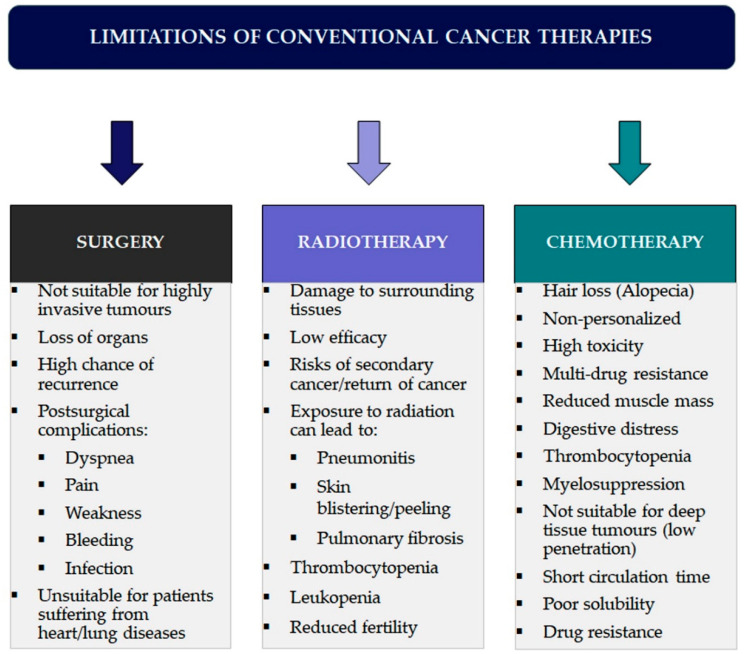
Overview of limitations and side effects associated with conventional cancer therapies.

**Figure 3 pharmaceuticals-17-00677-f003:**
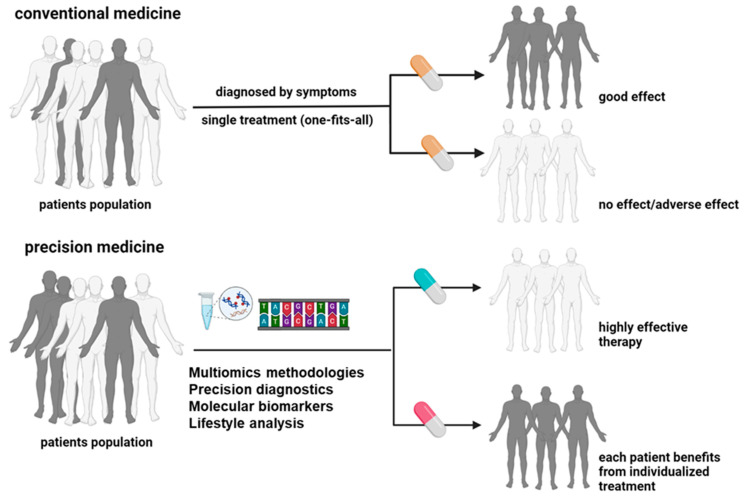
Conventional and precision medicine (PM) approaches have critical differences in classification factors, diagnostics methods, and treatment outcomes (created with BioRender.com).

**Figure 4 pharmaceuticals-17-00677-f004:**
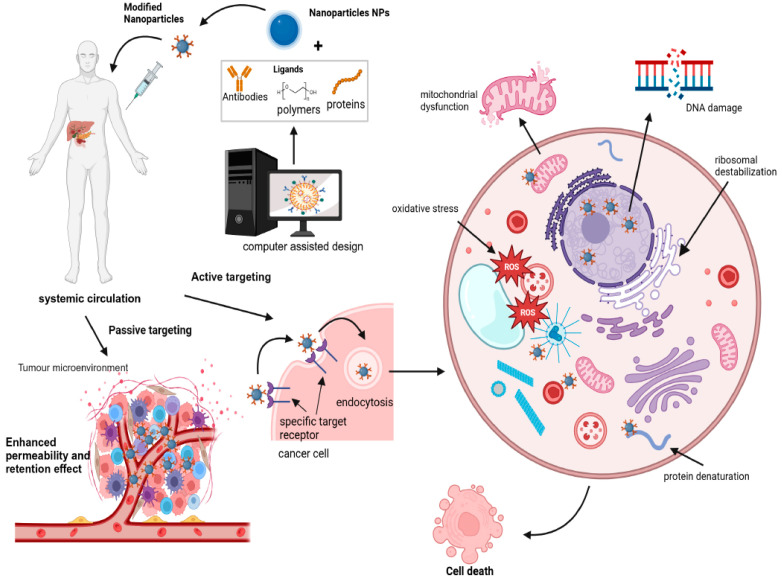
Conceptual illustration of nanoparticle-mediated targeted drug delivery in precision cancer medicine. Nanoparticles are engineered to carry therapeutic agents directly to cancer cells, enhancing drug efficacy and minimizing off-target effects. This approach leverages the unique properties of nanoparticles, such as their size, surface modifications, and responsiveness to specific biological stimuli, to improve treatment outcomes and reduce systemic toxicity (created with BioRender.com).

**Figure 5 pharmaceuticals-17-00677-f005:**
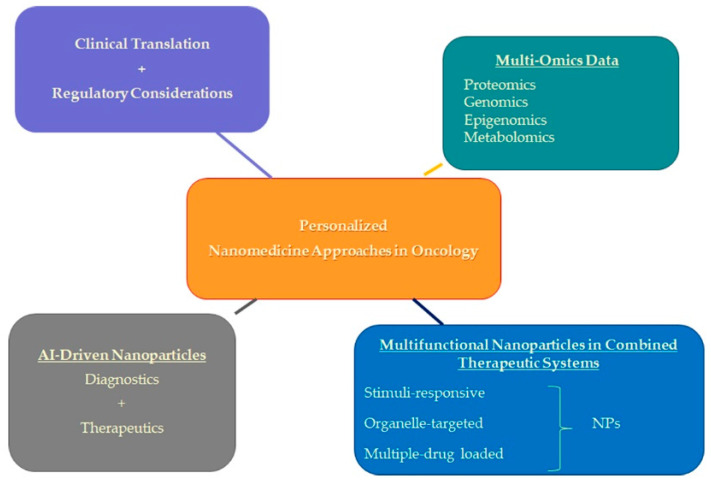
Future directions in personalized nanomedicine approaches in oncology.

**Table 2 pharmaceuticals-17-00677-t002:** Ligand-targeted nanoparticles for precision cancer therapy in clinical trials.

Clinical Name	Ligand	Receptor	Type NPs	Type of Cancer	Therapeutic Agent	Phase	ClinicalTrials.gov ID
Erbitux-EDVPAC	mAb	EGFR	EDV	Colorectal	Cetuximab	II	NCT00122460
MBP-426	Tf	TfR	Liposome	Adenocarcinoma	Leucovin/Fluorouracil	II	NCT00964080
NAb	mAB	HER2	Albumin Bound (NAB)	Breast cancer	Paclitaxel/Trastuzumab/Cyclophosphamide	II	NCT00629499
2B3-101	Glutathione	Glutathione transporters	Liposome	Leptomeningeal metastases	Doxorubicin	II	NCT01818713
MM-302	mAB	EGFR	Liposome	Breast cancer	Trastuzumab	I	NCT04622319
SGT-53	mAB	TfR	Liposome	Glioblastoma/Pancreatic cancer/Recurrent tumours	Temozolomide/p53 Gene Therapy(plasmid)	II	NCT02340156
BIND-014	Small molecules Ligands	PSMA	Polymeric	Carcinomas	Docetaxel	II	NCT01812746
EGFR(V)-EDV-Dox	BsAB	EGFR	Nanocell	Glioblastoma	Doxorubicin	I	NCT02766699
MesomiR 1	BsAB	EGFR	EDVs (non-living bacteria mini cells)	Malignant pleural mesothelioma/Lung cancer	miR-16	I	S0923753420307407
Intravenous EGFR-ErbituxEDVsMIT	BsAB	EGFR	EDVs (non-living bacteria living cells)	Paediatric solid and CNS tumours	Mitoxantrone packaged EDV	I	L01FE01-ATC code/EMA
SGT-94	TfRscFv	TfR	Liposome	Solid tumour	RB94 gene	I	NCT01517464

**Table 3 pharmaceuticals-17-00677-t003:** Summary of targeted formulations in clinical trials by cancer therapy categories.

Formulation	Target	Clinical Trial Phase	Cancer Type	Results	Category	Ref.
BIND-014	PSMA receptor	Phase I and II	Prostate cancer	Demonstrated potential in selectively targeting metastatic prostate tumour cells overexpressing PSMA receptors. Showed anti-tumour activity and increased median overall survival time.	Molecular and Ligand-Based Targeting	[94]
MM-302	HER2 receptor	Phase I and II	Metastatic breast cancer	It showed improved efficacy and safety profile in early trials, but the phase II trial (HERMIONE) did not show a clear benefit in terms of survival rate.	Molecular and Ligand-Based Targeting	[96]
PSMA-targeted polymeric nanoparticles	PSMA receptor	Phase I	Prostate cancer	Retained in the vascular compartment, it suggested improved efficacy in various cancer types and increased uptake by cancer cells/neovasculature.	Molecular and Ligand-Based Targeting	[94]
Rapamycin-PLGA nanoparticles	EGFR antibodies	Preclinical	Breast cancer (MCF-7 cells)	Showed increased cellular absorption and apoptotic activity in breast cancer cells.	Molecular and Ligand-Based Targeting	[97]
Human serum albumin NPs	Transferrin receptors	Preclinical	Breast cancer	Demonstrated potential to pass the blood-brain barrier and deliver medication to the intended location.	Molecular and Ligand-Based Targeting	[98]
Gelatin carriers functionalized with EGF	LHRHR receptors	Preclinical	Lung cancer	Functionalized carriers showed targeted cisplatin delivery to metastatic lung cancer cells and reduced kidney toxicity in mice.	Exploiting the Tumour Microenvironment	[103]
Mesoporous silica nanocarriers	LHRHR peptide	Preclinical	Lung cancer	Showed effective delivery of anti-cancer payloads into lung cancer cells and improved cytotoxicity.	Exploiting the Tumour Microenvironment	[104]
Polymer-based nanoparticles	CD44 marker	Preclinical	Lung cancer	Conjugated with hyaluronic acid facilitated lung cancer cell recognition and improved cytotoxicity.	Exploiting the Tumour Microenvironment	[105]
Acetylated dextran nanocarriers	iRGD peptide	Preclinical	Lung cancer	Engineered with iRGD and loaded with paclitaxel for targeted lung cancer treatment.	Exploiting the Tumour Microenvironment	[108]
Polyethylene glycol and chitosan nanocarriers	Folate receptor-alpha	Preclinical	Lung cancer	Targeted specifically folate receptor-alpha overexpressed lung tumour cells and demonstrated prolonged retention of paclitaxel inside the lungs.	Exploiting the Tumour Microenvironment	[109]
EGF-functionalized PLGA NPs	CD44 marker	Preclinical	Colon cancer	Inhibited colon tumour growth.	Exploiting the Tumour Microenvironment	[110]
Composite PLGA/PLA-PEG-FA NPs	Folate receptor	Preclinical	Colon cancer	Demonstrated suppression of colon tumour growth.	Exploiting the Tumour Microenvironment	[113]
PHBV/PLGA NPs	NA	Preclinical	Colon cancer	Exhibited promise as a nano-drug delivery system for colon cancer treatment.	Exploiting the Tumour Microenvironment	[114,115,116]

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
