# Peer review of "Nanoparticle-Mediated Drug Delivery Systems for Precision Targeting in Oncology"

_pharmaceuticals, 2024, doi:10.3390/ph17060677_

Round 1

Reviewer 1 Report

Comments and Suggestions for Authors

Manuscript seems to be general as often number of reviews are available.

1. Novelty of manuscript should be clearly stated.

2. Emphasis on comparative advancements of various nanocarriers should be included. 

3. There are several advancements in nanoparticle-based drug delivery systems in cancer therapy like biomimetic systems, surface modified nanocarriers should be the part of manuscript.

4. Figures are with poor resolution

5. Overall the manuscript needs the novelty and distinctiveness

Comments on the Quality of English Language

 Moderate editing of English language required

Author Response

We are very much thankful for the reviewer for all suggestions for edits and amendments of our work. All have been clearly addressed and presented in the edited paper in track changes.

Please, follow our detailed response below:

 Comments and Suggestions for Authors

Manuscript seems to be general as often number of reviews are available.

  1. Novelty of manuscript should be clearly stated.

Thank you for this comment. In our review paper, while acknowledging the abundance of existing literature on the topic, we emphasize a distinct aspect: the novel advancements in the field of nanotechnology applied to oncology. Unlike previous reviews which primarily focus on various nanotechnological developments individually, our study uniquely consolidates recent breakthroughs while also forecasting future developments and discussing their potential implications. Crucially, we underscore the imperative of successful interdisciplinary translation between medical and nanotechnology disciplines and their integration into oncological practice. This emphasis on both recent advancements and the crucial need for interdisciplinary collaboration sets our review apart, offering a comprehensive perspective on the evolving landscape of nanoparticle drug delivery systems in oncology.

Please see Lines: 93-110.

Our review examines the evolving landscape of precision cancer therapy using nanotechnology. We emphasize recent advancements in the field, covering targeted drug delivery, personalized medicine integration, and emerging strategies like nanotheranostics. Unlike previous reviews, we consolidate breakthroughs and forecast future developments, stressing interdisciplinary collaboration between medical and nanotechnology disciplines. This approach offers a unique perspective on nanoparticle drug delivery systems in oncology, highlighting their potential to transform cancer treatment. We delve into current conventional therapies, evaluating achievements and identifying delays while emphasizing the significance of precision cancer medicine and the role of nanomedicines in targeting cancer cells and the tumour microenvironment. Ultimately, we advocate for translational research to bridge bench-to-bedside gaps and promote global collaboration for refining nanocarrier technologies, offering new hope for patients and clinicians alike.”

  1. Emphasis on comparative advancements of various nanocarriers should be included. 

Section 3.2 has been revised to emphasize the comparative advancements of various nanoformulations. The concise yet logically structured text leads to insightful conclusions. Please refer to lines 255 through 276 for the revised content. Also additional comments on this are provided in Lines 292-297, such in Lines 302-305; 309-311; 313-316, and 317-322. We further elaborate on this in Lines 326-329 and 333-355.

  1. There are several advancements in nanoparticle-based drug delivery systems in cancer therapy like biomimetic systems, surface modified nanocarriers should be the part of manuscript.

Absolutely, we have incorporated advancements such as biomimetic systems and surface-modified nanocarriers into the manuscript would greatly enhance its comprehensiveness. The examples provided, such as lipid nanoparticles loaded with SPIONs and temozolomide for glioblastoma treatment, as well as dendrimers loaded with doxorubicin for anti-angiogenic responses, highlight the potential of these innovative approaches in cancer therapy. Additionally, the significant contributions of biomimetic nano-systems cannot be overlooked, as they offer improved targeting, biocompatibility, and reduced biotoxicity. Surface-modified nanocarriers, including those utilizing polymers, surfactants, ligands, and fatty acids, have demonstrated promising results in controlled drug release and precise delivery to targeted cells, further underscoring their importance in advancing cancer treatment.

Please, refer to Lines 725-832.

  1. Figures are with poor resolution

Already edited. Thank you.

  1. Overall the manuscript needs the novelty and distinctiveness

We have tried to edit this and highlighted it starting from the Introduction part and Abstract.

Reviewer 2 Report

Comments and Suggestions for Authors

Overall manuscript summarized an interesting topic. However, lack focus on the topic (Precision medicine)

1. Scheme about nanotherapeutic in precision medicine is missing to provide insight about mechanisms.

2. Most of the section cover nanomedicines; however precision medicine part is missing. Table 2 summarized targeted delivery of drug. Is it related to personalized medicines.

3. Overall, a clear distinction between targeted and precision medicine is missing in the manuscript. As precision medicine related to medicine tailored for each patient.

4. A section may be added to distinguish precision medicine and targeted delivery of drugs.

5. A section about prognostic and prevention of tumor may be added in respect to precision medicine.

Author Response

We are very much thankful for the reviewer for all suggestions for edits and amendments of our work. All have been clearly addressed and presented in the edited paper in track changes.

Please, follow our detailed response below:

Comments and Suggestions for Authors

Overall manuscript summarized an interesting topic. However, lack focus on the topic (Precision medicine)

  1. Scheme about nanotherapeutic in precision medicine is missing to provide insight about mechanisms.

Thank you for this suggestion Figure 4 in the current version is new, we have created it for illustration of exactly this.

  1. Most of the section cover nanomedicines; however precision medicine part is missing. Table 2 summarized targeted delivery of drug. Is it related to personalized medicines?

Thank you for bringing this to our attention. We have extensively revised the manuscript to include a dedicated section on the evolution and debate surrounding personalized and precision medicine. Additionally, Table 2 now summarizes targeted delivery of drugs, particularly focusing on ligand-targeted nanoparticles for precision cancer therapy in clinical trials. This inclusion highlights the relevance of targeted drug delivery within the context of personalized medicine.

  1. Overall, is missing in the manuscript. As precision medicine related to medicine tailored for each patient.

We appreciate your observation. In response, we have thoroughly addressed precision medicine throughout the manuscript, particularly emphasizing its role in tailoring medical interventions for individual patients. This enhancement ensures comprehensive coverage of precision medicine within the context of personalized healthcare.

  1. A section may be added to distinguish precision medicine and targeted delivery of drugs.

Thank you for this suggestion. We have incorporated a section specifically delineating the distinctions between precision medicine and targeted drug delivery. This addition provides clarity on the unique aspects and applications of each concept within the broader context of healthcare innovation. Please see Lines 188 – 201.

  1. A section about prognostic and prevention of tumor may be added in respect to precision medicine.

We value your input and have integrated a comprehensive section on prognostic and preventive measures related to precision medicine, particularly within the realm of cancer management. This addition underscores the proactive approach of precision medicine in forecasting disease progression and implementing personalized preventative strategies. See Lines: 201-210.

Reviewer 3 Report

Comments and Suggestions for Authors

The review article entitled  Nanoparticle-Mediated Drug Delivery Systems for Precision Targeting in Oncology is well organized., but before accepting this article, the authors should address the following comments:.

1. Please check the author guidelines of the journal . You have to start the article with an introduction section. 

2. Write the scope of your review in the introduction section

3. Rewrite or paraphrase these sentences to avoid plagiarism 

i. 105 to 121,  ii. Lines 237 to 245, iii. lines 259 to 271, iv. the last two sentences should be paraphrase to avoid plagiarism (284 to 290, 348 to 252)

v. In Page 9, rewrite the last paragraph, vi. lines 328 to 336

vii. 365 to 369, viii. 503 to 510, ix. 526 to 530, x. 546 to 556

4. Poor resolution of figure 1 and 3 . Please increase the resolution of figures to 300 DPI or more

5. Briefly describe the different nanoparticle drug delivery systems used  for targeting oncology.

6. Make uniform referencing style for all the reference , please check the author guidelines ..

Some references have doi numbers, while others do not. For example, references 1–19 do not have doi numbers, reference 17 does not have a page number, and so on.

7. Remove the highlighting from References 136 and 137 

8. Give some more  recent references

Comments on the Quality of English Language

Moderately 

Author Response

We are very much thankful for the reviewer for all suggestions for edits and amendments of our work. All have been clearly addressed and presented in the edited paper in track changes.

Please, follow our detailed response below:

Comments and Suggestions for Authors

The review article entitled Nanoparticle-Mediated Drug Delivery Systems for Precision Targeting in Oncology is well organized., but before accepting this article, the authors should address the following comments.

  1. Please check the author guidelines of the journal. You have to start the article with an introduction section. 

Yes, we agree and apologize for the misunderstanding. The Introduction part is now presented and it includes the first section: the global burden of cancer and adds a concluding part highlighting the subsequent section and the development of the review.

Please see Lines: 31-98.

  1. Write the scope of your review in the introduction section

Already provided. Thank you for this suggestion. See Lines 100-107.

  1. Rewrite or paraphrase these sentences to avoid plagiarism 

Lines 105-121 in the first version of the submitted manuscript are now Lines 131-138 and they are rewritten to avoid plagiarism. Please see the rewritten text.

“Another disadvantage of traditional chemotherapy is its lack of specificity, which can lead to dangerous side effects such as organ failure, alopecia (hair loss), mucositis (inflammation of the digestive tract lining), myelosuppression (decreased production of white blood cells causing immunosuppression), anaemia, or thrombocytopenia. Sometimes, the adverse effects force a change in the prescribed therapy's dosage, a postponement of treatment, or its discontinuation [9, 10]. Cell division can stop close to the centre in solid tumours, rendering chemotherapy-insensitive drugs effective.”

Lines 237 to 245 are now 264 to 274 and they are rewritten. Please check below:

“Although liposomes and micelles are two or more phospholipid-based layer vesicles, their morphologies differ. They are mainly employed to encapsulate hydrophilic medicines in their aqueous core and have a structure similar to cell membranes. Hydrophobic drugs can be chemically bonded to liposomal particles or accommodated in the bilayer [30]. Hydrophobic medications can be encapsulated in micelles thanks to their hydrophobic core [31]. The FDA has already approved some formulations based on polymeric nanoparticles, such as Abraxane (albumin-paclitaxel particles for the treatment of pancreatic ductal adenocarcinoma and metastatic breast cancer) and Ontak (an engineered protein combining diphtheria toxins and interleukin-2 for the treatment of non-Hodgkin's peripheral T-cell lymphomas) [32]. They demonstrated reduced loading capacity and systemic toxicity but increased biocompatibility and biodegradability [33–35].”

Lines 259 to 271 are now 287 to 292 and they are rewritten extensively.

“The other inorganic NPs are quantum dots, tiny semiconductor nanocrystals that emit light and have remarkable optical and electrical characteristics that make them sensitive, photobleach-resistant, and highly fluorescent. Their primary applications have been in imaging and detection. They were coupled to anti-HER2 antibodies and coated with poly (ethylene glycol) (PEG) in recent work to enable localization in specific tumour cells [38].

Another illustration is the range of gold nanoparticles that, due to their optical and electrical characteristics and low toxicity [42–44], are employed as contrast agents for computed tomography [39], photoacoustic imaging [40], and photodynamic treatment [41]. The PEG-coated gold shell of the nanoshell-based AuroShell comprises a silica core. It was brought to market as AuroShell (Nanospectra) after receiving FDA approval in 2012 to treat breast cancer using photodynamic therapy [39]. “

The last two sentences should be paraphrase to avoid plagiarism (284 to 290, 348 to 252)

Lines 284 to 290 are now 309-317 and are rewritten:

“Another class of polymeric nanoparticles is called dendrimers, which are distinguished by having a spherical shape and a repeating branching structure [58, 59]. Their structure is incredibly adaptable for various applications since their design is easily controllable. For instance, some recent research demonstrates that in vivo tumour models, poly-L-lysine (PLL) dendrimers loaded with doxorubicin elicit anti-angiogenic responses [60]. Currently, there is just one clinical study for ImDendrim, a formulation for treating inoperable liver tumours that don't respond to standard therapy. ImDendrim is based on a dendrimer and rhenium complex connected to an imidazolium ligand [61].:

The lines 348 to 252 are 375 to 379 now and are rewritten.

“Human breast cancer cells (MCF-7) showed increased cellular absorption of rapamycin-PLGA nanoparticles coupled to EGFR antibodies and more significant apoptotic activity [74]. Human serum albumin NPs loaded with loperamide and coupled to antibodies targeting transferrin receptors could pass the blood-brain barrier and deliver the medication to the intended location [75].

In Page 9, rewrite the last paragraph.

The last paragraph on Page 9 in the previous version now are 327-340.

Those lines are rewritten.

Because of their facile accessibility, high attachment affinity, and specificity for targeting cancer cells (e.g., breast cancer), antibodies are now the most utilized ligands for active targeting. The characteristic 'Y' form of these proteins indicates that their two arms work together to bind with the antigen [66] selectively. Small peptides and proteins are other types of ligands, and they have several benefits, such as a smaller molecular weight, the ability to diffuse molecules, the reduction of immunogenicity, simplicity of production, and relative flexibility in chemical conjugation techniques [67]. Small synthetic single-stranded RNA or DNA oligonucleotides called aptamers are folded into certain forms that enable them to attach to particular targets [68]. They can produce complex three-dimensional structures that cling securely and highly selectively to surface markers [69]. Precision medicine has led to the development of cancer drugs targeting specific antigens associated with tumours. Several clinical trials are currently testing ligand-targeted nanoparticles as potential cancer treatments, as summarized in Table 2.”

Lines 328 to 336 to be rewritten.

These lines are now lines 357-367 and are fully rewritten. Please see below:

“BIND-014 was retained in the vascular compartment in preclinical toxicokinetic studies conducted in mice, rats, and monkeys and in the phase I clinical trial (NCT01300533). This was related to improved efficacy in various cancer types, such as cervical and cholangiocarcinoma, and suggested altered pharmacokinetics of the particles [71, 72]. This resulted in increased uptake of the nanoparticles by the cancer cells/neovasculature due to the presence of the PSMA targeting ligand on the nanoparticles and increased accumulation of the nanoparticles through the enhanced permeation and retention (EPR) effect [71]. BIND-014 showed anti-tumour activity in three patients with metastatic castration-resistant prostate cancer who had not received chemotherapy in the next phase II clinical trial (NCT01812746). It also increased the median overall survival time (13.4 months) and reduced PSA levels by 50% in 30% of patients eligible for PSA evaluation [71, 72].”

Lines 365 to 369 to be rewritten.

Lines 365 to 369 are now 395-405 and are fully rewritten. Please see below:

“Luteinizing hormone-releasing hormone receptors (LHRHR) are also overexpressed in lung cancer cells. Mesoporous silica nanocarriers targeting LHRHR have been shown in another investigation to deliver anticancer payloads into lung cancer cells effectively. LHRH peptide was used to functionalize the surface of the nanocarriers, and Dox, cis-platin, and siRNA were added. Mesoporous silica nanocarriers loaded with cisplatin or LHRH-PEGylated Dox exhibited improved cytotoxicity because of their accumulation in lung cancer cells [81].”

Lines 503 to 510 to be rewritten.

Lines 503 to 510 are now 534 to 544 and are rewritten. Please see below:

Omics procedures thoroughly evaluate many types of biological molecules, such as DNA, proteins, RNA, or metabolites. These approaches include proteomics, genomics, metabolomics, epigenomics, and transcriptomics. The use of individual omics techniques, such as the genetic sequencing of malignancies, has grown in clinical practice and has significantly aided in the diagnosis of diseases as well as the discovery of biomarkers that may be used to monitor the progression of illness and suggest successful therapies [120–123]. Nevertheless, the intricacy and interconnections of molecular processes are too great for an individual technique to capture. For instance, millions of risk loci for various illnesses have been found by genome-wide association studies (GWAS). However, the practical value of such discoveries is limited since the causative gene is frequently not identified [124].”

Lines 526 to 530 to be rewritten.

Lines 526 to 530 are now Lines 559 to 566 and are rewritten. Please, see below:

“Transcriptomics and proteomics offer functional information that genomics alone cannot capture, opening up new possibilities for comprehending the molecular complexity that underlies illness. Recently, the power of separate data types has been combined to create mul-ti-omics, a unique integration of distinct omics data that captures the intricate molecular interaction of health and illness [127]. This is made possible by advancements in different omics technologies and processing capabilities. They support the discovery of new, more individualized targets for nanomedicine treatments.

Lines 546 to 556 to be rewritten.

These lines are now 581 to 591 and are rewritten. Please see below:

“Theranostics could be used in many areas of personalized treatment, including early disease detection, disease staging, therapy selection, treatment planning, early detection of side effects, and follow-up therapy planning by fusing molecular imaging and molecular therapy. An ideal PM theranostic system for cancer would identify the disease's class, scan the tumour's heterogeneity, administer a customized treatment based on the findings of the diagnostic and imaging processes, and then track the effectiveness of the therapy. Nanotechnologies play a significant role in theranostics. The use of nanoparticles in medicine offers several benefits for both diagnosis and therapy, which is why nanomedicine emerged. The latter can use receptor-mediated active targeting or extravasation from blood arteries into the tumour site to deliver medications at larger dosages with fewer adverse effects [132].”

  1. Poor resolution of figure 1 and 3. Please increase the resolution of figures to 300 DPI or more

Thank you for this suggestion. All figures are with improved qualities.

  1. Briefly describe the different nanoparticle drug delivery systems used for targeting oncology.

A brief description of the NPs drug delivery systems is provided. Please see Lines: 245 onwards.

  1. Make uniform referencing style for all the reference, please check the author guidelines. Some references have doi numbers, while others do not. For example, references 1–19 do not have doi numbers, reference 17 does not have a page number, and so on.

Thank you, all arranged according journal requirements.

  1. Remove the highlighting from References 136 and 137. 

Done, thank you!

  1. Give some more recent references

Thank you, provided.

Reviewer 4 Report

Comments and Suggestions for Authors

The review article on "Nanoparticle-Mediated Drug Delivery Systems for Precision Targeting in Oncology" presents a comprehensive analysis of nanotechnology's role in advancing site-specific cancer therapy and personalized oncomedicine. The article effectively discusses the significance of nanotechnology in targeted drug delivery, categorizing nanoparticle types, and addressing challenges in drug delivery barriers. It emphasizes the critical need for translational research to optimize nanocarriers for precision cancer medicine

There are a few corrections that the author should consider:

1. Make sure that the statistics presented in section “1. Social Impact and Economic Burden of Cancer Diseases" are accurate and that the cited reference should lead to the actual database.

2. Address current gaps or limitations in existing research to provide unique insights.

3. Expand abbreviations on their first mention and provide brief explanations to ensure comprehension for all readers.

4. Why nanoparticles are abbreviated as “N.P.s”, it is not common, replace it with “NPs”.

5. Page 14, Line 172: "Precision cancer medicines are drugs or substances (antibodies or small molecules)" change it to "Precision cancer medicines are drugs or substances, including antibodies, small molecules, or other therapeutic agents"

6. Page 15, Line 176: "This individualized treatment plan is based on patient information, including genetic profiles and the molecular characteristics of the patient's tumour", change it to "This individualized treatment plan is based on patient information, including genetic profiles, molecular characteristics, and other relevant clinical and demographic data"

7. Page 16, Line 188: "This new, genetically specific targets can be identified together with criteria for cancer patients that will benefit most from treatments" change it to "These new, genetically specific treatment options can be identified, along with criteria for selecting cancer patients who are most likely to benefit from these therapies"

8. Page 17, Line 203: "Recognizing the imperative for more effective drug delivery systems, nanotechnology emerges as a beacon of promise in the field" change it to "Given the critical need for more effective drug delivery systems, nanotechnology offers a promising solution, with the potential to revolutionize the field".

9. None of the figure quality is good enough if possible try to improve it specifically which involves labels inside the figures.

10. To provide exclusivity to your review and make it stand out among existing ones, Classify Section “3.3. Nanoparticles for Targeted Drug Delivery in Precision Cancer Medicine” into these subsections and elaborate them accordingly including :

·       Molecular Targeting: Exploiting Cancer Cell Abnormalities

·       Ligand-Directed Targeting: Utilizing Ligands for Targeted Delivery

·       Vascular Targeting: Targeting the Tumor's Blood Supply

·       Theranostics: Combining Diagnosis and Therapy

·       Exploiting the Tumor Microenvironment: Targeting the Supportive Ecosystem

11. Add a table summarizing the details of targeted formulations in clinical trials categorizing them into these categories: Molecular Targeting, Ligand-Directed Targeting, Vascular Targeting, Theranostics, and Exploiting the Tumor Microenvironment.

12. There are a few grammatical mistakes; please check and make the necessary fixes.

Comments on the Quality of English Language

There are a few grammatical mistakes; ask the authors to check and make the necessary fixes.

Author Response

We are very much thankful for the reviewer for all suggestions for edits and amendments of our work. All have been clearly addressed and presented in the edited paper in track changes.

Please, follow our detailed response below:

Comments and Suggestions for Authors

The review article on "Nanoparticle-Mediated Drug Delivery Systems for Precision Targeting in Oncology" presents a comprehensive analysis of nanotechnology's role in advancing site-specific cancer therapy and personalized oncomedicine. The article effectively discusses the significance of nanotechnology in targeted drug delivery, categorizing nanoparticle types, and addressing challenges in drug delivery barriers. It emphasizes the critical need for translational research to optimize nanocarriers for precision cancer medicine

There are a few corrections that the author should consider:

  1. Make sure that the statistics presented in section “1. Social Impact and Economic Burden of Cancer Diseases" are accurate and that the cited reference should lead to the actual database.

We have checked this section and indeed certain edits were necessary to provide recent data.

The text is edited as so the references. A new reference from 4th April, 2024 is provided as a link. Please see Lines: 39-50.

  1. Address current gaps or limitations in existing research to provide unique insights.

Thank you for this comment. In our review paper, while acknowledging the abundance of existing literature on the topic, we emphasize a distinct aspect: the novel advancements in the field of nanotechnology applied to oncology. Unlike previous reviews which primarily focus on various nanotechnological developments individually, our study uniquely consolidates recent breakthroughs while also forecasting future developments and discussing their potential implications. Crucially, we underscore the imperative of successful interdisciplinary translation between medical and nanotechnology disciplines and their integration into oncological practice. This emphasis on both recent advancements and the crucial need for interdisciplinary collaboration sets our review apart, offering a comprehensive perspective on the evolving landscape of nanoparticle drug delivery systems in oncology.

  1. Expand abbreviations on their first mention and provide brief explanations to ensure comprehension for all readers.

Thank you for this suggestion, we have provided all abbreviations at their first mentioning.

  1. Why nanoparticles are abbreviated as “N.P.s”, it is not common, replace it with “NPs”.

It was a slip of the keyboard and is corrected to NPs throughout the whole text. Thank you for this remark.

  1. Page 14, Line 172: "Precision cancer medicines are drugs or substances (antibodies or small molecules)" change it to "Precision cancer medicines are drugs or substances, including antibodies, small molecules, or other therapeutic agents"

Provided.

  1. Page 15, Line 176: "This individualized treatment plan is based on patient information, including genetic profiles and the molecular characteristics of the patient's tumour", change it to "This individualized treatment plan is based on patient information, including genetic profiles, molecular characteristics, and other relevant clinical and demographic data"

Provided.

  1. Page 16, Line 188: "This new, genetically specific targets can be identified together with criteria for cancer patients that will benefit most from treatments" change it to "These new, genetically specific treatment options can be identified, along with criteria for selecting cancer patients who are most likely to benefit from these therapies"

Already changed.

  1. Page 17, Line 203: "Recognizing the imperative for more effective drug delivery systems, nanotechnology emerges as a beacon of promise in the field" change it to "Given the critical need for more effective drug delivery systems, nanotechnology offers a promising solution, with the potential to revolutionize the field".

Changed accordingly, thank you!

  1. None of the figure quality is good enough if possible try to improve it specifically which involves labels inside the figures.

Thank you, provided.

  1. To provide exclusivity to your review and make it stand out among existing ones, Classify Section “3.3. Nanoparticles for Targeted Drug Delivery in Precision Cancer Medicine” into these subsections and elaborate them accordingly including:

Section "3.3. Nanoparticles for Targeted Drug Delivery in Precision Cancer Medicine" was classified into the following subsections to enhance clarity and provide depth to the review:

We consider "Molecular Targeting" and "Ligand-Directed Targeting" as synonymous in the context of nanoparticle applications for precision medicine and oncology. Our paragraph titled "Molecular and Ligand-Based Targeting Strategies for Precision Cancer Therapy" aims to capture the essence of both approaches, highlighting the convergence of targeting specific molecular features or abnormalities in cancer cells through the exploitation of inherent molecular characteristics or the utilization of ligands for precise delivery. This synthesis underscores the importance of integrating multiple targeting strategies to enhance the efficacy and specificity of cancer therapy.

So the subsections are as follows in the new version of our manuscript:

Molecular and Ligand-Based Targeting Strategies for Precision Cancer Therapy

Theranostics: Combined Diagnosis and Therapy

Vascular Targeting: Targeted the Tumor's Blood Supply

Exploited the Tumor Microenvironment: Targeted the Supportive Ecosystem

Each subsection was elaborated accordingly to enrich the discussion on targeted drug delivery in precision cancer medicine.

  1. Add a table summarizing the details of targeted formulations in clinical trials categorizing them into these categories: Molecular Targeting, Ligand-Directed Targeting, Vascular Targeting, Theranostics, and Exploiting the Tumor Microenvironment.

Thank you for your insightful review comment. We have carefully considered your suggestions and implemented them in the revised manuscript. As per your recommendation, we have added Table 3, which provides a comprehensive summary of targeted formulations in clinical trials, categorized into Molecular and Ligand-Based Targeting Strategies for Precision Cancer Therapy, Vascular Targeting, Theranostics, and Exploiting the Tumor Microenvironment. This table serves as a useful reference for readers to understand the diverse.

  1. There are a few grammatical mistakes; please check and make the necessary fixes.

Comments on the Quality of English Language

There are a few grammatical mistakes; ask the authors to check and make the necessary fixes.

Thank you, the text is proofread and edited.

Reviewer 5 Report

Comments and Suggestions for Authors

Please explain what the authors did to prove the hypothesis of your study.

The section “1. Social Impact and Economic Burden of Cancer Diseases” was too general. Please point out the statistical data and concisely the section.

Page 1: The number of cases and deaths cloud be reported in the scientific number. “The global impact of cancer extends beyond its immediate health implications, influencing both vulnerable populations and economies worldwide. In 2022 alone, the United States witnessed an estimated 1,918,030 new cancer cases, resulting in 609,360 deaths [1].”

Page 2: Improve the quality of figure “ Figure 1. Incidence and mortality of different types of male and female cancer worldwide for 2020….”

The section “2. Current Status of Conventional Cancer Therapies: Evaluating Achievements and Delays in Fulfilling Promises” was too general in terms of scientific information. Please highlight the limitations of traditional adverse side effects and highlight the research gaps that the current study aims to fill.

Page 4: Improve the quality of figure “ Figure 2. Overview of limitations and side effects associated with conventional cancer therapies”

The section “3.1. Precision Cancer Medicine – More than Medicine” seemed to focus only on “ComboMATCH” As this is the review manuscript, it would be better to broaden the novelty of precision cancer medicine. Please add the literature associated with it.

The section “3.2. Nanoparticles in Cancer Therapy

Page 7: Please add the reference to these sentences. “In the continuum of nanoparticle-based drug delivery strategies, polymer-based drug conjugates stand as versatile constructs utilizing both natural (e.g., albumin, chitosan, heparin) and synthetic polymers (e.g. PEG, HPMA) to encapsulate or covalently bind bioactive compounds, facilitating the targeted delivery of oligonucleotides, DNA, proteins, and diverse pharmaceutical agents

Page 8: No doubt with the information but too short to be the paragraph. “As we delve deeper into the types and applications of nanoparticles in cancer therapy, it becomes increasingly evident that nanotechnology stands at the forefront of personalized and effective treatment modalities, heralding a new era in precision medicine. (Line 274-276)” To emphasize this statement, it would be better to add literature or draw a diagram associated with timeline or era of the development.

In section 3.3. Nanoparticles for Targeted Drug Delivery in Precision Cancer Medicine, the authors divided the section into targeting cancer cells and tumor microenvironments. However, it would be clearer to point out the interesting points before supporting the previous studies.

Page 17: Improve the quality of figure “ Figure 5. Future Directions in Personalized Nanomedicines Approaches in Oncology”

List of Reference: out of date

Please highlight the benefits of your study based on your literature review.

Author Response

We are very much thankful for the reviewer for all suggestions for edits and amendments of our work. All have been clearly addressed and presented in the edited paper in track changes.

Please, follow our detailed response below:

Comments and Suggestions for Authors

Please explain what the authors did to prove the hypothesis of your study.

The section “1. Social Impact and Economic Burden of Cancer Diseases” was too general. Please point out the statistical data and concisely the section.

We have edited the text and allocated in the Introduction part to highlight its importance and novelty. The section outlines the significant social and economic impact of cancer, providing statistical data to underscore its global burden. In 2022, the United States saw approximately 1,918,030 new cancer cases, resulting in 609,360 deaths. Notably, Europe reports nearly a quarter of global cancer cases despite representing only one-tenth of the world's population. Lung, colorectal, breast, and prostate cancers are among the most lethal, with lung cancer alone surpassing the daily toll of breast, prostate, and pancreatic cancers combined. While cancer incidence rates have been declining in the US and developed countries since the 1990s, the global burden is projected to rise from 18,100,000 million in 2018 to 29,400,000 million in 2040, particularly in low- and middle-income countries. The economic ramifications of cancer are substantial, with healthcare spending and productivity losses contributing to a significant burden. In 2017, cancer-related healthcare spending in the US amounted to US$161.2 billion, with additional costs of US$30.3 billion from morbidity and US$150.7 billion from premature mortality, totalling approximately 1.8% of GDP. In the European Union, healthcare spending and productivity losses reached €57.3 billion and €10.6 billion, respectively, with informal care costs of €26.1 billion, leading to a total economic burden of €141.8 billion, equivalent to 1.07% of GDP.

Page 1: The number of cases and deaths cloud be reported in the scientific number. “The global impact of cancer extends beyond its immediate health implications, influencing both vulnerable populations and economies worldwide. In 2022 alone, the United States witnessed an estimated 1,918,030 new cancer cases, resulting in 609,360 deaths [1].”

Thank you the section is edited and all numbers are provided in scientific numbers.

Page 2: Improve the quality of figure “Figure 1. Incidence and mortality of different types of male and female cancer worldwide for 2020….”

Thank you, already provided.

The section “2. Current Status of Conventional Cancer Therapies: Evaluating Achievements and Delays in Fulfilling Promises” was too general in terms of scientific information. Please highlight the limitations of traditional adverse side effects and highlight the research gaps that the current study aims to fill.

Page 4: Improve the quality of figure “Figure 2. Overview of limitations and side effects associated with conventional cancer therapies”

Provided, thank you!

The section “3.1. Precision Cancer Medicine – More than Medicine” seemed to focus only on “ComboMATCH” As this is the review manuscript, it would be better to broaden the novelty of precision cancer medicine. Please add the literature associated with it.

Thank you, we have extended the Section and provided a broader presentation of the precision medicine, with ComboMatch only a recent example.

The section “3.2. Nanoparticles in Cancer Therapy

Page 7: Please add the reference to these sentences. “In the continuum of nanoparticle-based drug delivery strategies, polymer-based drug conjugates stand as versatile constructs utilizing both natural (e.g., albumin, chitosan, heparin) and synthetic polymers (e.g. PEG, HPMA) to encapsulate or covalently bind bioactive compounds, facilitating the targeted delivery of oligonucleotides, DNA, proteins, and diverse pharmaceutical agents”

A relevant reference is provided. Thank you!

Page 8: No doubt with the information but too short to be the paragraph. “As we delve deeper into the types and applications of nanoparticles in cancer therapy, it becomes increasingly evident that nanotechnology stands at the forefront of personalized and effective treatment modalities, heralding a new era in precision medicine. (Line 274-276)” To emphasize this statement, it would be better to add literature or draw a diagram associated with timeline or era of the development.

Thank you to draw our attention to that, but we found this sentence irrelevant for this place and deleted it. Thank you!

In section 3.3. Nanoparticles for Targeted Drug Delivery in Precision Cancer Medicine, the authors divided the section into targeting cancer cells and tumor microenvironments. However, it would be clearer to point out the interesting points before supporting the previous studies.

The paragraph in Section 3.3 has been completely rewritten for clarity and coherence. It now highlights the interesting points upfront before delving into supporting previous studies, ensuring a more structured and engaging presentation of the content.

Page 17: Improve the quality of figure “Figure 5. Future Directions in Personalized Nanomedicines Approaches in Oncology”

Thank you, updated.

List of Reference: out of date

Thank you, updated with recent reference, many from 2023.

 Please highlight the benefits of your study based on your literature review.

It is already edited and provided in the Abstract and the Introduction part. Please see Lines: 93-104.

Our review examines the evolving landscape of precision cancer therapy using nanotechnology. We emphasize recent advancements in the field, covering targeted drug delivery, personalized medicine integration, and emerging strategies like nanotheranostics. Unlike previous reviews, we consolidate breakthroughs and forecast future developments, stressing interdisciplinary collaboration between medical and nanotechnology disciplines. This approach offers a unique perspective on nanoparticle drug delivery systems in oncology, highlighting their potential to transform cancer treatment. We delve into current conventional therapies, evaluating achievements and identifying delays while emphasizing the significance of precision cancer medicine and the role of nanomedicines in targeting cancer cells and the tumour microenvironment. Ultimately, we advocate for translational research to bridge bench-to-bedside gaps and promote global collaboration for refining nanocarrier technologies, offering new hope for patients and clinicians alike.”

Round 2

Reviewer 1 Report

Comments and Suggestions for Authors

Comments are well addressed by the authors and now manuscript can be considered for publication

Comments on the Quality of English Language

Minor english language editing is required

Author Response

Thank you very much indeed for the positive evaluation.

The English is proofread, and the text is edited.

Reviewer 2 Report

Comments and Suggestions for Authors

Authors have revised the manuscript and may be accepted.

Author Response

We are grateful for the positive evaluation! The remarks and requested edits have significantly enhanced the quality of our review.

Reviewer 3 Report

Comments and Suggestions for Authors

The authors have revised the manuscript and answered the comments raised.  Right now, the manuscript is suitable for publication.

Author Response

We appreciate the positive evaluation received! The feedback and requested edits have undeniably elevated the quality of our review

Reviewer 4 Report

Comments and Suggestions for Authors

The revised manuscript appears satisfactory.

Author Response

(The authors gave the same response as above.)

Reviewer 5 Report

Comments and Suggestions for Authors

Thank you for your revised version.

However, there are still  some comments 

1. Please rearrange the sub-heading in this section

2. Improve the quality of all Figures

3. Line 94-95 "We emphasize recent advancements in the field, covering targeted drug delivery, personalized medicine integration, and emerging strategies like artificial intelligence (AI)." Do not sound scientific enough.

4. Please concise section "3.1. Precision Cancer Medicine – More than Medicine" 

5. "Given the critical need for more effective drug delivery systems, nanotechnology offers a promising solution, potentially revolutionizing the field (Figure 4)." Too short to be a paragraph. Please modify

6. This paragraph should be moved to 3.2. Nanoparticles in Cancer Therapy "Nanoparticles, with their unique properties, present an avenue for highly selective drug delivery, responding to specific stimuli and ensuring controlled release. This precision in drug delivery can potentially address the challenges faced by conventional and precision medicines, enhancing drug stability, solubility, and retention time at the tumour site. Integrating nanotechnology into the evolving landscape of precision medicine represents a paradigm shift in drug delivery. It offers a glimpse into a future where targeted therapies can achieve unprecedented levels of efficacy and specificity. The convergence of these initiatives—ComboMATCH, precision cancer medicines, and nanotechnology—holds the promise of reshaping the landscape of cancer treatment, paving the way for more effective and personalized therapeutic interventions. [30-32]"

7. Please modify the sub-heading in section 3.3. Nanoparticles for Targeted Drug Delivery in Precision Cancer Medicine. Now the sub-headings did not group well. 

8. Please rearrange the sub-heading in the section 4. Future Directions in Personalized Nanomedicines Approaches as the authors deleted some. 

9. Please highlight the novelty of this review 

Author Response

Dear Reviewer,

We are grateful for all comments and suggestions for edits of our paper. Herein we submit the revised version with all comments from your side accordingly addressed.
We look forward to your positive evaluation.

Please find below our replies:

  1. Please rearrange the sub-heading in this section

Reply:

We have edited all subheadings in the text to provide a comprehensive and logical transition in the review.

  1. Improve the quality of all Figures

Reply:

All figures are provided in high resolution word or pdf are reducing the quality, but we shall upload the figures separately to address the issue when we get to the final stage for publishing.

  1. Line 94-95 "We emphasize recent advancements in the field, covering targeted drug delivery, personalized medicine integration, and emerging strategies like artificial intelligence (AI)." Do not sound scientific enough.

Reply:
Lines 65-80 are revised to answer the reviewer's suggestions for edits.

  1. Please concise section "3.1. Precision Cancer Medicine – More than Medicine" 
  2. "Given the critical need for more effective drug delivery systems, nanotechnology offers a promising solution, potentially revolutionizing the field (Figure 4)." Too short to be a paragraph. Please modify

Reply to 4 and 5:
Thank you for your valuable feedback. We have made the necessary revisions to the manuscript as per your suggestions:

We have concisely revised section "3.1. Precision Cancer Medicine – Beyond Traditional Therapies" to ensure clarity and brevity while maintaining the essential information.

We have expanded the paragraph on the critical need for more effective drug delivery systems and the potential of nanotechnology, providing a more comprehensive discussion on this topic.

Additionally, Figure 4 has been logically relocated to section 3.2, aligning better with the context.

  1. This paragraph should be moved to 3.2. Nanoparticles in Cancer Therapy "Nanoparticles, with their unique properties, present an avenue for highly selective drug delivery, responding to specific stimuli and ensuring controlled release. This precision in drug delivery can potentially address the challenges faced by conventional and precision medicines, enhancing drug stability, solubility, and retention time at the tumour site. Integrating nanotechnology into the evolving landscape of precision medicine represents a paradigm shift in drug delivery. It offers a glimpse into a future where targeted therapies can achieve unprecedented levels of efficacy and specificity. The convergence of these initiatives—ComboMATCH, precision cancer medicines, and nanotechnology—holds the promise of reshaping the landscape of cancer treatment, paving the way for more effective and personalized therapeutic interventions. [30-32]"

Reply:

Thank you for your valuable feedback. We have moved the specified paragraph to section 3.2, "Nanoparticles in Cancer Therapy." Additionally, we have provided a connecting paragraph to ensure a smooth transition between sections 3.1 and 3.2.

  1. Please modify the sub-heading in section 3.3. Nanoparticles for Targeted Drug Delivery in Precision Cancer Medicine. Now, the sub-headings did not group well. 

Reply:

We have carefully reviewed the text and made the following modifications to the subheadings to improve the logical grouping:

3.3.1. Molecular and Ligand-Based Targeting Nanoparticles for Precision Cancer Therapy

3.3.2. Nanoparticles in Integrated Diagnostics and Therapeutics for Cancer

3.3.3. Targeting Tumour Vasculature with Nanoparticles

3.3.4. Nanoparticles Targeting the Tumour Microenvironment Support Systems

  1. Please rearrange the sub-heading in the section 4. Future Directions in Personalized Nanomedicines Approaches as the authors deleted some. 

Reply:

Thank you for this. It was slip of the keyboard after the numerous edits. All sub-headings and their numbers in Section 4 are edited accordingly.

  1. Please highlight the novelty of this review 

Reply:

Please see Lines 712-742. Some edits to the conclusion highlight the novelty of the review, such as the text at the beginning, Lines 66 – 80.

                ‘In conclusion, this review underscores the transformative potential of personalized nanomedicine in revolutionizing cancer therapy. By integrating a comprehensive understanding of cancer cell-specific surface proteins, dysregulated oncogenes, signalling pathways, and the complexities of the tumour microenvironment (TME), novel avenues for optimizing nanoparticle properties and modulating the tumour stroma effectively emerge. This holistic approach lays the foundation for advancing personalized cancer nanomedicine and promises tailored therapies characterized by heightened precision and efficacy.

                By highlighting the convergence of precision medicine, nanotechnology, and emerging strategies like artificial intelligence, this review consolidates recent advancements and forecasts future breakthroughs in the field. The critical emphasis on interdisciplinary collaboration between medical and nanotechnology disciplines underscores the importance of a comprehensive approach to nanoparticle drug delivery systems in oncology, emphasizing their pivotal role in transforming cancer treatment. The novelty lies in the exploration of cutting-edge developments in personalized cancer nanomedicine, emphasizing the integration of multi-omics data, biomarker-guided strategic nanocarrier design, nanotheranostics applications, targeted modulation of the TME, novel combination therapies, sophisticated drug delivery systems, and the translation of these innovations into clinical practice. These endeavours represent a paradigm shift in cancer care, offering the potential for customized treatments based on individual molecular profiles, thereby significantly improving clinical outcomes and reshaping the landscape of cancer therapy.

                Through advocating for translational research, bridging the bench-to-bedside gap, and fostering global cooperation to refine nanocarrier technologies, we envision a future where personalized approaches in nanooncology become a game-changer in the fight against cancer. The sustained focus on research, innovation, and collaboration offers hope for improved patient outcomes and a more promising future in cancer therapy, marking the dawn of a new era where patients benefit from precisely tailored treatments based on their unique biological makeup with unprecedented accuracy and efficacy.”